# Revisiting Mixout: An Overlooked Path to Robust Finetuning

## Abstract

Finetuning vision foundation models often improves in-domain accuracy but comes at the cost of robustness under distribution shift. We revisit Mixout, a stochastic regularizer that intermittently replaces finetuned weights with their pretrained reference, through the lens of a single-run, weight-sharing implicit ensemble. This perspective reveals three key levers that govern robustness: the *masking anchor*, *resampling frequency*, and *mask sparsity*. Guided by this analysis, we introduce GMixout, which (i) replaces the fixed anchor with an exponential moving-average snapshot that adapts during training, and (ii) regulates masking period via an explicit resampling-frequency hyperparameter. Our sparse-kernel implementation updates only a small fraction of parameters with no inference-time overhead, enabling training on consumer-grade GPUs. Experiments on benchmarks covering covariate shift, corruption, and class imbalance, ImageNet / ImageNet-LT, DomainNet, iWildCam, and CIFAR100-C, GMixout consistently improves in-domain accuracy beyond zero-shot performance while surpassing both Model Soups and strong parameter-efficient finetuning baselines under distribution shift.

## 1 Introduction

A core assumption in standard machine/deep learning is that training and test examples are independently and identically distributed (i.i.d.) (Vapnik, 1991). However, in practice, real-world datasets often violate this assumption due to shifts in input marginal distributions (Koh et al., 2021; Gulrajani & Lopez-Paz, 2021), spurious correlations between inputs and labels (Geirhos et al., 2020), and imbalanced class distributions (Yang et al., 2023; Liu et al., 2019). The advent of foundation models opens new opportunities for addressing distribution shifts (Bommasani, 2021; Radford et al., 2021; Oquab et al., 2023). These models exhibit strong transferability across tasks, lowering data requirements while improving robustness and generalization. Notably, they often perform well even in zero-shot settings (Radford et al., 2021), underscoring how large-scale, heterogeneous pretraining can alleviate many challenges. Nonetheless, in most applications, domain-specific data is available and can be leveraged to further improve performance on distributions closer to the target (Tian et al., 2023; Wortsman et al., 2022b).

Robust finetuning aims to achieve competitive in-domain (ID) performance while maintaining the out-of-distribution (OOD) robustness of a pretrained model when transferring it to a downstream task. However, prior strategies have notable limitations: (i) domain-invariant representation learning, while theoretically appealing, can be overly restrictive and often underperforms strong zero-shot baselines (Arjovsky et al., 2019; Sun & Saenko, 2016; Ganin et al., 2016); (ii) augmentation-based methods depend on domain expertise to craft invariance-preserving transforms (Gontijo-Lopes et al., 2020; Liu et al., 2022); (iii) ensembling and model soups (multiple independent models with aggregated predictions or weights) can be effective but are prohibitively expensive at foundation-model scale (Wortsman et al., 2022a; Rame et al., 2022); and (iv) parameter-efficient finetuning (PEFT) methods—such as Random Masking (Xu & Zhang, 2024) or low-rank adaptation (LoRA) (Hu et al., 2022) that update only a small fraction of parameters—still degrade when the test distribution drifts even modestly from the finetuning data (Shuttleworth et al., 2024; Han et al., 2024; Hu et al., 2022; Xu & Zhang, 2024).

We argue that a practical and robust finetuning method should satisfy four criteria: (1) preserve generalization to unseen but related distributions, (2) improve downstream accuracy over the zero-

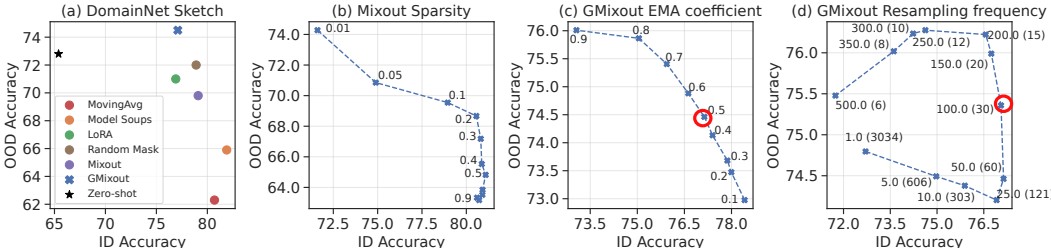

Figure 1: **OOD-ID accuracy trade-off on the DomainNet dataset. Models are trained on Sketch (ID) and evaluated on Real, Painting, and Clipart (OOD) data.** (a) Each point represents a method, with ID accuracy on the x-axis and OOD accuracy on the y-axis. The ideal method should improve over the zero-shot baseline along both axes. (b) Mixout regulates the ID–OOD trade-off through mask sparsity. (c, d) GMixout extends Mixout with two mechanisms: (i) an EMA coefficient that updates the masking anchor, and (ii) a resampling frequency that determines the number of uncovered subnetworks during optimization (shown in parentheses in (d)). By controlling the variance and covariance of the expected test error, these hyperparameters significantly enhance OOD performance while maintaining competitiveness on ID. Although these additional hyperparameters can significantly affect the OOD trade-off, they remain stable across datasets, and all results are reported using the same settings across experiments (indicated by red circles in the plots).

shot model, (3) incur no additional cost at inference, and (4) enable training on consumer-grade GPUs. As shown in Figure 1 (a) on the DomainNet benchmark—where models are trained on the Sketch domain (ID) and evaluated on the remaining three domains of Real, Painting and Clipart (OOD)—existing baselines often satisfy some of the desired criteria but still degrade under even mild distribution shifts. In particular, although methods such as LoRA and Random Masking can substantially improve downstream accuracy, their OOD performance still lags behind that of the zero-shot model.

Motivated by these gaps, we revisit Mixout (Lee et al., 2019) as a robust PEFT mechanism. Mixout applies a binary mask to randomly replace finetuned weights with their frozen pretrained counterparts, regularizing learning to stay close to the initialization along the optimization path (Lee et al., 2019). As shown in Figure 1 (b), mask sparsity provides a knob to balance ID–OOD performance. We argue, however, that the strength of Mixout comes from a deeper principle: it implicitly ensembles random subnetworks with extensive weight sharing during training. Like other ensemble methods, its test error depends on the uncorrelated diversity of ensemble members (subnetworks), reducing variance and improving generalization (Dietterich, 2000; Allen-Zhu & Li, 2023). We formalize this view by decomposing the expected test error of Mixout into bias–variance–covariance–locality (BVCL) terms (Rame et al., 2023). This analysis highlights two overlooked levers beyond sparsity: (i) the anchor weights to which Mixout reverts, and (ii) the frequency of mask resampling. In its original form, Mixout resamples at every step and always reverts to the initial pretrained weights—choices that increase subnetwork correlation and restrict downstream adaptation. To address these limitations, we propose GMixout, which (a) replaces the fixed anchor with an exponential moving-average (EMA) snapshot that evolves during training, and (b) introduces a resampling-frequency hyperparameter to control how often weight switching occurs. The EMA coefficient balances adaptability to the downstream task against robustness from the pretrained checkpoint (Figure 1(c)), while suitable resampling frequencies—controlling the number of distinct subnetworks optimized—consistently improve OOD accuracy without reducing ID performance (Figure 1(d)). Importantly, these hyperparameters are stable in practice, and all reported results use the same settings across experiments. Finally, we provide an efficient implementation of GMixout using sparse CUDA kernels (Gale et al., 2020), enabling finetuning of large foundation models on consumer-grade GPUs.

In summary, our main contributions are threefold: **(1)** We present an ensemble-based perspective on the effectiveness of Mixout and identify three key hyperparameters that govern Mixout generalization. **(2)** Guided by this analysis, we introduce *GMixout*, which (i) replaces the fixed pretrained source with an exponential moving average snapshot, (ii) introduces a refresh-frequency hyperparameter, and (iii) incorporates a memory- and compute-efficient implementation using sparse

GPU kernels, enabling scalable training without storing dense masks. **(3)** We validate GMixout across diverse out-of-distribution benchmarks, including: covariate shift (ImageNet, DomainNet, iWildCam), common corruptions (CIFAR100-C), and class-imbalance (ImageNet-LT, CIFAR100-LT). GMixout consistently outperforms strong baselines under domain shifts, while meeting all four practicality criteria.

## 2 RELATED WORKS

**Robust finetuning.** Deep models often degrade sharply under distribution shift. Domain Generalization (DG) addresses this by training on multiple sources to generalize to unseen domains (Blanchard et al., 2011; Wang et al., 2022), but practical assumptions (e.g., access to meta-information) limit applicability (Pezeshki et al., 2024), and DG methods are often outperformed by large pretrained models using even simple strategies (Koh et al., 2021; Gulrajani & Lopez-Paz, 2021). Recent advances in foundation models based on vision–language models (Radford et al., 2021; Zhai et al., 2023; Jia et al., 2021) and self-supervised learning (Oquab et al., 2023), trained on massive heterogeneous datasets, have further highlighted this robustness, with zero-shot and lightweight adaptations exceeding the performance of prior DG approaches (Radford et al., 2021; Addepalli et al., 2024). Building on this trend, Goyal et al. (2023); Oh et al. (2024) show that mimicking the pretraining loss during finetuning consistently outperforms standard finetuning strategies, though their methods are limited to vision–language models. More general approaches constrain finetuned weights from drifting too far from pretrained ones (Kumar et al., 2022; Xuhong et al., 2018; Tian et al., 2023; 2024). For example, Lee et al. (2019) shows that Mixout acts as an adaptive weight decay that mitigates overfitting on limited data. Ensembling approaches like Model Soups further improves robustness (Wortsman et al., 2022a; Arpit et al., 2022; Rame et al., 2023; Wenzel et al., 2020), as diversity across models reduces variance (Dietterich, 2000; Rame et al., 2022; Allen-Zhu & Li, 2023). Yet training ensembles are costly, requiring multiple finetuned models. Efficient variants exploit weight-space geometry (Jang et al., 2025), moving averages (Izmailov et al., 2018; Tarvainen & Valpola, 2017), weight-averaging with the original model (Wortsman et al., 2022b), dynamic weight-averaging at test time (Oh et al., 2025; Zhu et al., 2024)or low-rank factorizations (Wen et al., 2020). However, these still trade off ensemble diversity or require full-parameter updates—untenable at foundation-model scale (Bommasani, 2021).

**Parameter-efficient finetuning methods.** PEFT reduces adaptation cost by updating only a small subset of parameters (Han et al., 2024). Notable techniques that can match the performance of full finetuning include Adapters (Houlsby et al., 2019; Chen et al., 2022), which insert bottleneck modules into transformer blocks, and Low-Rank Adaptation (LoRA) (Hu et al., 2022; Liu et al., 2024), which approximates weight updates with low-rank matrices. Compared to adapter-based approaches, the LoRA family has the advantage of incurring no additional inference overhead. More recently, Xu & Zhang (2024) introduced a method that applies a fixed random binary mask to model parameters, updating only the unmasked subset. This approach achieves performance on par with, or even exceeding, LoRA while still leveraging the full expressive capacity of the pretrained model. Despite these advances, PEFT methods typically yield only modest improvements in out-of-distribution robustness relative to the zero-shot model (Biderman et al., 2024; Shuttleworth et al., 2024). A related line of work augments the prompt with trainable continuous vectors, known as prompt tuning (Li et al., 2025; Zhou et al., 2022b). These methods are tailored to vision–language foundation models and are most effective in few-shot learning settings (Zhou et al., 2022a). In contrast, our approach is a general finetuning strategy that scales to medium- and large-scale downstream datasets and applies equally well beyond the vision–language setting (Further comparisons are provided in Appendix A.3).

## 3 BACKGROUND AND UNIFIED FINETUNING FORMULATION

**Task and evaluation.** Given training data from $P_{\mathrm{id}}$, our goal is to learn a predictor $f : \mathbb{R}^d \to \mathcal{Y}$ that generalizes to test samples from $P_{\mathrm{ood}}$. Since $P_{\mathrm{ood}}$ is unavailable, we minimize the empirical risk over the training data. For a loss function $\ell$, this is defined as:

$$\mathcal{L}(f, \Phi) \;=\; \mathbb{E}_{(\boldsymbol{x}, y) \sim P_{\mathrm{id}}} \big[ \ell \big( f(x), y \big) \big],$$

where $\Phi$ is the parameter of the predictor. In practice, $P_{\text{ood}}$ may differ from $P_{\text{id}}$ through (1) covariate shift in $P(x)$, (2) class imbalance in $P(y)$, or (3) changes in $P(y|x)$ due to spurious correlations absent at test time.

**Models.** In this work, we focus on predictors that leverage pretrained representations. We instantiate the final predictor $f$ as follows: given features $h_\theta(x) \in \mathbb{R}^k$ from a feature extractor with parameters $\theta$, and a classification head $g_w : \mathbb{R}^k \to \mathcal{Y}$ with parameters $w$. In our case, the feature extractor is CLIP's vision encoder (Radford et al., 2021) implemented with a ViT (Dosovitskiy et al., 2020). For the classification head $g$, a prototypical head is adopted with $\ell_2$-normalized features and class prototypes, and apply a temperature scaling on the logits. The parameters $w$ are initialized via the CLIP text encoder. The full set of parameters is denoted as $\Phi = \{\theta, w\}$. Although our main experiments use CLIP as the base model, our method is compatible with any vision encoder (see Section 5.2).

### 3.1 A UNIFIED PARAMETERIZATION OF FINETUNING

Let $\Phi_0 = (\theta_0, w_0)$ denote the pretrained initialization, and let $\Delta$ be a zero-initialized learnable parameter of the same shape as $\Phi_0$, representing the weight residual relative to the pretrained initialization. Standard full-parameter finetuning of $f$ on $P_{\text{id}}$ can be formulated as:

$$\min_\Delta \mathcal{L}(f, \Phi_0 + \Delta). \tag{1}$$

In practice, full-parameter finetuning requires excessive GPU memory, especially for tuning foundation models using consumer-grade GPUs (Han et al., 2024). To address this, LoRA decomposes the weight update $\Delta$ into low-rank matrices as follows:

$$\Delta = \alpha A B, \quad A \in \mathbb{R}^{m \times r}, \ B \in \mathbb{R}^{r \times n}, \tag{2}$$

where $m$ and $n$ are the dimensions of $\Delta$, and $\alpha \in \mathbb{R}$ is a fixed scaling factor, typically set to the inverse of the rank parameter $r$.

As LoRA approximates $\Delta$ using low-rank matrices, it may limit the expressive capacity of pretrained models (Shuttleworth et al., 2024; Xu & Zhang, 2024). Random Masking (Xu & Zhang, 2024; Sung et al., 2021) applies a binary mask $\mathbf{M}$ to weight updates $\Delta$, freezing masked elements and optimizing only the unmasked ones. Each $\mathbf{M}$ is sampled i.i.d. from Bernoulli$(1 - p)$, where $p$ is the masking probability and sparsity is $s = 1 - p$. Masks are generated once at initialization and fixed during finetuning. The corresponding risk minimization is:

$$\min_\Delta \mathcal{L}(f, \Phi_0 + \mathbf{M} \odot \Delta), \tag{3}$$

where $\odot$ denotes the Hadamard product.

Similar to Random Masking, Mixout also applies a masking operation; however, Mixout resamples the elements of $\mathbf{M}$ at every iteration. To ensure that the expected output of a neuron matches the actual output at test time, Mixout rescales $\Delta$ by $(1 - p)^{-1}$ during training. The corresponding objective is:

$$\min_\Delta \mathbb{E}\big[\mathcal{L}\big(\Phi_0 + \tfrac{1}{1-p}(\mathbf{M} \odot \Delta)\big)\big], \tag{4}$$

where the expectation is taken over the random masks $\mathbf{M}$.

### 3.2 A SIMPLE REGULARIZATION VIEW FOR MASKED UPDATES

Assume that the loss function $\mathcal{L}$ is strongly convex in a neighborhood of $\Phi_0$ and consider the Mixout parameterization as in equation 4. Lee et al. (2019) by a first-order expansion around $\Phi_0$ shows the following inequality:

$$\mathbb{E}\big[\mathcal{L}\big(\Phi_0 + \tfrac{1}{1-p}(M \odot \Delta)\big)\big] \ \geq \ \mathcal{L}(\Phi_0) \ + \ \frac{\mu\,(p)}{2(1-p)} \, \|\Delta\|_2^2, \tag{5}$$

where $\mu > 0$. Equation 5 shows that, as the mask probability $p$ increases, the effective $\ell_2$-penalty grows, biasing the solution toward $\Phi_0$.

## 4 FROM MIXOUT TO GMIXOUT

GMixout generalizes Mixout in two ways. First, it introduces a hyperparameter $k$ that controls the frequency of mask resampling (with Mixout corresponding to $k = 1$). We define an *episode*, indexed by $i$, as the $k$ optimization steps performed with a fixed random mask. Second, GMixout updates the pretrained anchor $\Phi_0$ using the exponential moving average of $\Delta_i$ and the previous anchor $\Phi_{i-1}$, after which $\Delta$ is reset to zero. This modification effectively replaces the fixed anchor with a moving-average version of the pretrained weights, which then serves as the basis for subsequent updates. The risk minimization under GMixout is given by:

$$\min_{\Delta} \mathbb{E}\big[\mathcal{L}\big(\Phi_i + \tfrac{1}{1-p}(\mathbf{M} \odot \Delta)\big)\big], \quad \Phi_i = \lambda\Phi_{i-1} + (1 - \lambda)\Delta. \tag{6}$$

In practice, $k$ depends on the total number of training iterations. For clarity, we express it in terms of $I$, the total number of episodes in GMixout. The modifications required to obtain GMixout from Mixout are summarized in the Algorithm 1.

**Implementation Notes.** For Random Masking, Mixout and GMixout, storing dense masks is inefficient. Instead, we keep $(\mathbf{S}, \Delta_{\mathbf{S}})$ for the unmasked indices $\mathbf{S}$ and reconstruct $\mathbf{M} \odot \Delta$ with sparse CUDA kernels (Gale et al., 2020). At inference, all methods merge $\Delta$ into $\Phi_0$, adding no extra cost.

---

**Algorithm 1** GMixout and Mixout Training Procedure

1: Initialize model parameters $\Phi_0$, mask probability $p$, total number of episodes $I$, EMA coefficient $\lambda$, total training iterations $T$.

Define $\Delta \leftarrow 0$, $k \leftarrow 1$ or $k \leftarrow \lfloor \frac{T}{I} \rfloor$

2: **for** each iteration $iter$ **do**
3:     **if** $iter \bmod k = 0$ **then**
4:         $\Phi_i \leftarrow \lambda\Phi_{i-1} + (1 - \lambda)\Delta$
5:         $\Delta \leftarrow 0$
6:         $\mathbf{M} \sim \text{Bernoulli}(1 - p)$
7:     **end if**
8:     Update parameters according to Eq. 4 or Eq. 6.
9: **end for**

---

### 4.1 GMIXOUT ERROR ANALYSIS AND CONNECTION TO ENSEMBLING

An intuitive way to understand GMixout is through the lens of ensembling in weight space. Specifically, consider $M$ finetuned models, each trained under an identically distributed (i.d.) learning procedure $l_{P_{\text{id}}}$. The expected error of a model with weights $\Phi_{\text{WA}} = \frac{1}{M} \sum_{m=1}^{M} \Phi_m$ with respect to the joint distribution $L_{P_{\text{id}}}^M = \{l_{P_{\text{id}}}^{(m)}\}_{m=1}^M$ can be decomposed into (Rame et al., 2022)[1]:

$$\mathbb{E}_{L_{P_{\text{id}}}^M}\big[\mathcal{L}_{\text{ood}}(\Phi_{WA})\big] = \mathbb{E}_{(x,y)\sim P_{\text{ood}}}\Big[\text{bias}^2(x,y) + \frac{1}{M}\text{var}(x) + \frac{M-1}{M}\text{cov}(x)\Big] + O(\bar{\Pi}^2), \tag{7}$$

where $\text{bias}(x,y)$ denotes the irreducible error of the predictor even with infinite data, $\text{var}(x)$ reflects variability due to the stochasticity of the learning procedure, and $\text{cov}(x)$ measures the prediction covariance between two ensemble members whose weights are averaged. The locality term $\bar{\Pi}^2$ quantifies the expected squared distance between individual weights and their average. Eq. 7 shows that ensembling $M$ models reduces the variance by a factor of $M$, but also introduces covariance and locality terms, which—together with bias—must be controlled to ensure low OOD error.

Now consider the GMixout training procedure. Following the above analysis, we can define the aggregated GMixout weights through the episodes as $\Phi_{\text{GMixout}} = \frac{1}{I} \sum_{i=1}^{I} \Phi_i$. In GMixout, the total number of episodes $I$ influences both variance and covariance: variance depends on the total number of optimized subnetworks, while covariance depends on how many optimization steps each subnetwork takes (fewer steps increase correlation, Figure 1(d)). The mask probability $p$ further affects

---

[1] We refer the reader to the Appendix A.1 for the detailed formulation.

covariance through the overlap of active coordinates across episodes. With small $p$, subnetworks collide more often, increasing covariance (Figure 1(b)). Finally, both $p$ and $\lambda$ jointly control the implicit $\ell_2$-penalty toward the pretrained checkpoint (Eq. 5), which keeps the locality term small and governs the trade-off between adaptability towards the downstream task against robustness from the pretrained checkpoint (Figure 1(b)). These findings are further corroborated in Section 5.2.

# 5 RESULTS AND DISCUSSION

**Models and Datasets.** By default, we use the ViT-B/16 architecture pretrained on OpenAI's CLIP dataset, unless stated otherwise. Our experiments cover a broad range of distribution shift scenarios. We evaluate on ImageNet-1K (Russakovsky et al., 2015), with OOD tests on ImageNet-V2 (Recht et al., 2019), ImageNet-R (Hendrycks et al., 2021a), ImageNet-Sketch (Wang et al., 2019), and ImageNet-A (Hendrycks et al., 2021b). From the WILDS benchmark (Koh et al., 2021), we include iWildCam (Beery et al., 2021), which contains animal images with variations in camera types, backgrounds, and illumination. We also consider DomainNet (Peng et al., 2019), where one domain is treated as in-distribution and the remaining three as OOD. To test robustness under synthetic corruptions, we use CIFAR100-C (Hendrycks & Dietterich, 2019), which introduces 15 corruption types with the highest severity (5). For long-tailed class imbalance, we evaluate on CIFAR100-LT (Cao et al., 2019) and ImageNet-LT (Liu et al., 2019). In both datasets, the number of training samples per class follows a long-tailed distribution that decreases exponentially, while the test set is balanced with a uniform distribution across classes.

**Baselines.** We compare GMixout to baselines including PEFT methods (LoRA, Random Mask), full-parameter finetuning, a moving-average variant with EMA coefficient 0.99, and Mixout. Additional baselines include Model Soups, constructed from five independently finetuned models varying in augmentation and batch order. These baselines represent state-of-the-art choices under reflective benchmarks—Model Soups for covariate shift and corruption (Rame et al., 2022; Arpit et al., 2022), and PEFT methods for imbalance recognition (Shi et al., 2024). Since our goal is to achieve strong performance on both ID and OOD data, we adopt relatively high ranks and masking ratios for LoRA and mask-based methods. Our initial experiments on two domains of DomainNet (Real and Sketch (see Section 5.2)) indicate that updating roughly $10\%$ of the parameters in the ViT-B/16 model is sufficient, which corresponds to LoRA with rank $r = 64$ and a masking ratio of $s = 0.1$. For GMixout, we set $\lambda = 0.5$ and choose $k$ so that the total number of uncovered subnetworks during optimization is approximately $I = 30$. In all cases, we replace every linear layer in the network with its respective modified counterpart. Additional details on hyperparameter selection are provided in the Appendix A.5.

## 5.1 MAIN RESULTS

Our experiments raise the following three major observations on GMixout. We first evaluate on medium-scale datasets (40k–250k samples): DomainNet, iWildCam, and CIFAR100 (Table 1). On DomainNet, the zero-shot CLIP baseline is already strong (70.1%), but GMixout achieves the best average accuracy (**72.3%**), surpassing zero-shot by **+2.3**, Model Soups by **+7.3**, and the best PEFT method by **+2.9** while leading on all domains (Real/Sketch/Painting/Clipart). On iWildCam, where zero-shot performance is very low (ID: 11.5, OOD: 12.7), GMixout delivers a large boost, attaining the best in-domain macro-F1 (**50.4**) and second-best OOD score (37.0), just **0.6** below Model Soups and ahead of all other PEFT baselines. On CIFAR100 and CIFAR100-C, where CLIP's low-resolution transfer is limited, GMixout remains competitive: 90.9% on clean (just **0.2** below Model Soups) and 62.1% under corruptions (**0.9** below). On long-tailed benchmarks (ImageNet-LT and CIFAR100-LT; Table 2), GMixout again sets the best overall results: **77.0** on ImageNet-LT and **82.2** on CIFAR100-LT. Gains are most pronounced in scarce-data regimes: it achieves the top scores on Medium (**76.2**, **82.5**) and Few-shot (**70.9**, **76.0**) categories, while staying competitive on Many-shot.

---

**Observation 1**

GMixout yields consistent gains across covariate shifts, long tails, and corruptions—often rivaling or surpassing Model Soups and strong PEFT methods.

---

Table 1: OOD accuracy of GMixout and SOTA methods on four DomainNet domains (train on one, evaluate on the others), macro-F1 on ID and OOD test sets of iWildCam (WILDS), and accuracy on CIFAR100 and CIFAR100-C (corruptions).

| Method | DomainNet (OOD) | | | | | iWildCam | | CIFAR100 | |
|---|---|---|---|---|---|---|---|---|---|
| | Real | Sketch | Painting | Clipart | Avg. | ID | OOD | ID | Corruption |
| Zero-shot | 65.8 | 72.8 | 71.4 | 70.4 | 70.1 | 11.5 | 12.7 | 64.3 | 33.3 |
| Full-FT | 60.2 | 62.4 | 61.6 | 61.9 | 61.5 | 46.5 | 36.4 | 89.7 | 57.6 |
| MovingAvg | 60.2 | 62.3 | 62.0 | 61.8 | 61.6 | 46.6 | 34.2 | 89.9 | 58.7 |
| Model Soups | 64.3 | 65.9 | 64.6 | 65.2 | 65.0 | 46.8 | **37.6** | **91.1** | **63.0** |
| Linear Probing | 62.2 | 64.4 | 60.2 | 62.7 | 62.4 | 31.6 | 24.7 | 74.1 | 42.9 |
| LoRA | 66.5 | 71.0 | 69.0 | 69.5 | 69.0 | 49.7 | 35.9 | 89.9 | 55.0 |
| Random Mask | 66.8 | 72.0 | 68.9 | 69.8 | 69.4 | 47.9 | 35.2 | 90.2 | 56.1 |
| Mixout | 66.5 | 69.8 | 65.0 | 67.7 | 67.3 | 43.5 | 32.9 | 88.7 | 55.7 |
| **GMixout (Ours)** | **68.0** | **74.5** | **73.5** | **73.4** | **72.3** | **50.4** | 37.0 | 90.9 | 62.1 |

Table 2: Balanced accuracy of GMixout and SOTA methods on the long-tailed benchmarks ImageNet-LT and CIFAR100-LT.

| Method | ImageNet-LT | | | | CIFAR100-LT | | | |
|---|---|---|---|---|---|---|---|---|
| | All | Many | Med. | Few | All | Many | Med. | Few |
| Zero-shot | 68.2 | 69.4 | 67.8 | 66.4 | 62.0 | 66.2 | 61.8 | 57.3 |
| Full-FT | 73.1 | 80.3 | 70.8 | 60.4 | 79.6 | 88.1 | 79.9 | 69.3 |
| MovingAvg | 73.0 | 79.0 | 71.0 | 63.1 | 81.0 | 89.4 | 80.6 | 71.9 |
| Model Soups | 76.0 | 81.5 | 74.5 | 65.5 | 82.1 | **89.9** | 82.2 | 73.0 |
| Linear Probing | 74.2 | 77.8 | 73.3 | 67.4 | 70.0 | 77.2 | 71.1 | 60.4 |
| LoRA | 76.7 | **81.6** | 75.3 | 67.4 | 81.1 | 85.7 | 81.0 | 75.7 |
| Random Mask | 76.8 | 80.7 | 75.5 | 70.4 | 81.5 | 87.4 | 81.7 | 74.6 |
| Mixout | 70.9 | 76.5 | 69.7 | 59.3 | 81.5 | 88.1 | 82.3 | 72.9 |
| **GMixout (Ours)** | **77.0** | 80.3 | **76.2** | **70.9** | **82.2** | 87.3 | **82.5** | **76.0** |

Next we present the results on a large-scale benchmark. Table 4 reports results on ImageNet-1K (ID) and five natural distribution shifts derived from ImageNet (OOD). As expected, methods that fine-tune on ImageNet-1K substantially improve ID accuracy over zero-shot CLIP: Full-FT, MovingAvg, LoRA, and GMixout all reach around 83–84% ID accuracy, compared to 68.2% for zero-shot.

However, the picture changes under a distribution shift. Zero-shot CLIP remains strong on IN-R (76.4) and competitive on IN-A (51.6), while full finetuning methods (Full-FT, MovingAvg), including Mixout, degrade to the mid-40s on IN-A. Model Soups partially restore robustness but require costly multi-model ensembles. PEFT methods offer the best trade-off: slightly lower ID accuracy than Model Soups but consistently higher OOD averages than zero-shot. Among them, GMixout achieves the highest OOD average (**60.7**), surpassing Model Soups (60.0), LoRA (59.9), and Mask (60.5), with strong results on IN-R (71.6) and IN-Sketch (50.4), while remaining competitive on IN-A (47.2).

Experiments on medium- and large-scale datasets show GMixout is most effective at medium scale, with its advantage persisting—though narrowing—at larger scales. Figure 2 illustrates this trend: across all ImageNet subsamples (25–200 shots per class) and the full dataset, GMixout maintains a strong ID–OOD balance. Unlike Model Soups, which favors ID accuracy, or MovingAvg, which fluctuates, PEFT methods—especially GMixout—scale smoothly with data. Even at 25 shots, GMixout maintains strong OOD accuracy, and with more data, it matches or surpasses the strongest alternatives.

> **Observation 2**
>
> As the number of training samples grows, GMixout's performance gap narrows but remains persistent.

Finally, table 3 summarizes training benchmarks across baselines, reporting trainable parameters, latency, memory, and FLOPs per gradient update with batch size 64 (details in Appendix A.4). Please note that although GMixout and Mixout have the same total number of trainable parameters as Full-FT, in our implementation only the unmasked parameters participate in backpropagation.

This substantially reduces the per-step computation and memory footprint, making GMixout feasible on limited hardware. PEFT methods have comparable FLOPs, with LoRA offering lower latency via better hardware utilization. On consumer GPUs, memory remains the main bottleneck, which GMixout alleviates through sparse kernels. All methods share the same inference cost, since $\Delta$ can be merged into $\Phi_0$ after training.

> **Observation 3**
>
> GMixout strikes an effective balance between accuracy and efficiency, maintaining competitive performance across diverse datasets while lowering compute and memory costs.

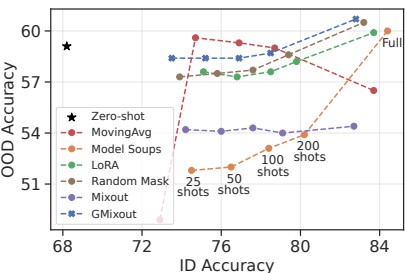

Figure 2: OOD-ID accuracy on ImageNet with varying training set sizes.

Table 3: Training benchmarks of GMixout and baselines for a single gradient update on batch size of 64.

| Method | Training (per mini-batch) | | | |
| --- | --- | --- | --- | --- |
| | #Params ↓ (M) | FLOPs ↓ (T) | Latency ↓ (ms) | GPU VRAM ↓ (G) |
| Full-FT | 85.5 | 1.7 | 70.4 | 3.2 |
| MovingAvg | 85.5 | 1.7 | 70.4 | 3.2 |
| Model Soups | 427.5 | 8.5 | 352.0 | 16.0 |
| Linear Probing | 0.5 | 0.6 | 22.5 | 0.5 |
| LoRA | 8.7 | 1.3 | 77.6 | 2.3 |
| Random Mask | 9.0 | 1.1 | 112.0 | 2.5 |
| Mixout[2] | 9.0 | 1.1 | 123.1 | 2.6 |
| **GMixout (Ours)** | 9.0 | 1.1 | 123.1 | 2.6 |

Table 4: Accuracy of GMixout and SOTA methods on ImageNet-1k (ID) and four natural distribution shifts from ImageNet-1k (OOD).

| Method | ID | OOD | | | | |
| --- | --- | --- | --- | --- | --- | --- |
| | IN-1k | IN-V2 | IN-R | IN-Sketch | IN-A | Avg. |
| Zero-shot | 68.2 | 62.0 | **76.4** | 46.6 | 51.6 | 59.1 |
| Full-FT | 83.7 | 73.9 | 65.0 | 47.1 | 40.0 | 56.5 |
| MovingAvg | 83.7 | 73.9 | 64.9 | 47.0 | 40.0 | 56.5 |
| Model Soups | **84.4** | **74.9** | 68.9 | **50.9** | 45.3 | 60.0 |
| Linear Probing | 77.0 | 66.4 | 70.0 | 43.7 | 45.7 | 56.5 |
| LoRA | 83.7 | 74.0 | 69.3 | 49.6 | 46.6 | 59.9 |
| Random Mask | 83.2 | 74.0 | 70.4 | 49.9 | **47.7** | 60.5 |
| Mixout | 82.7 | 72.6 | 62.7 | 45.0 | 37.4 | 54.4 |
| **GMixout (Ours)** | 82.8 | 73.6 | 71.6 | 50.4 | 47.2 | **60.7** |

## 5.2 ABLATIONS STUDIES

This section presents empirical analyses validating our theoretical insights, followed by studies on how the parameter ratio influences the ID–OOD trade-off and how base model choice impacts GMixout. We then report GMixout's performance on language understanding benchmarks. Unless otherwise specified, all vision ablations are conducted on DomainNet Real and Sketch using the same setup as the main experiments, with additional results provided in Appendix A.2.

**GMixout validation.** To further examine the impact of GMixout hyperparameters (EMA coefficient $\lambda$ and resampling frequency $k$) on the ID–OOD trade-off, we conduct the same analysis as in Figure 1(c,d) on DomainNet Real (ID), which differs markedly from Sketch, as evident in their zero-shot CLIP performance (65.2 vs. 72.8). In Fig. 4, we vary $\lambda$ for fixed $k$ and vice versa, with sparsity fixed at 0.1. Results show a clear trade-off for $\lambda$: larger values improve OOD accuracy at the expense of ID accuracy. By contrast, higher $k$, up until $500$, consistently improves both ID and OOD performance. Notably, the stable range of $k$ is broader on Sketch (50–300) (Figure 4(right)) than on Real (50–100) (Figure 1(d)).

---

[2]For Mixout, we report the numbers obtained using sparse-kernel implementation. The original Mixout implementation does not reduce the memory; it only zeroes out gradients during the backward pass but still must store all gradients, resulting in the same per-step cost as Full-FT.

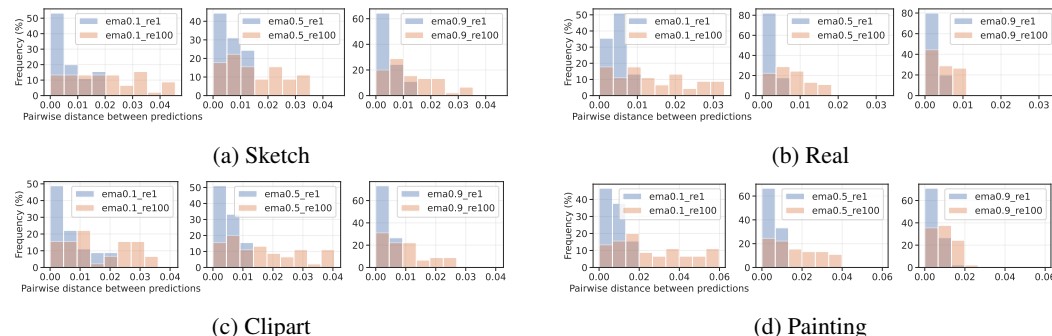

(a) Sketch                                     (b) Real

(c) Clipart                                   (d) Painting

Figure 3: Frequencies of prediction diversities (Aksela, 2003) between two subnetworks obtained from a single training run on DomainNet-Sketch, under $\lambda \in \{0.1, 0.5, 0.9\}$ and $k \in \{1, 100\}$. Across both in-domain (a) and out-of-domain (b, c, d) evaluations, setting the resampling parameter to $k = 1$ yields virtually no diversity for many of the checkpoint pairs. Increasing $k$ substantially boosts prediction diversity, as expected. With respect to the GMixout EMA strength $\lambda$, the highest diversity occurs at $\lambda = 0.1$, and diversity decreases monotonically as $\lambda$ increases.

**Prediction diversity.** Diversity among ensemble members is widely recognized as a key driver of ensemble effectiveness (Dietterich, 2000; Rame et al., 2022; Fort et al., 2019; Izmailov et al., 2018). In Figure 7, we validate that GMixout also benefits from such diversity. We quantify prediction diversity using the ratio-error metric (Aksela, 2003), defined as the ratio $N_{\text{diff}}/N_{\text{simul}}$ between the number of different errors $N_{\text{diff}}$ and simultaneous errors $N_{\text{simul}}$ on test samples for each pair of checkpoints obtained during GMixout training on DomainNet-Sketch, under $\lambda \in \{0.1, 0.5, 0.9\}$ and $k \in \{1, 100\}$. A higher average over the $\binom{M}{2}$ checkpoint pairs indicates that members are less likely to make the same errors. For $k = 100$, we extract the weights from the last ten re-masking steps of GMixout. To ensure fair comparison, we extract an equivalent number of checkpoints for the $k = 1$ setting over the same training interval. Across both in-domain evaluations (Figure 7a) and out-of-domain evaluations (Figures 7b, 7c, and 7d), the trend is consistent: when $k = 1$, prediction diversity is essentially zero for many of the checkpoint pairs, indicating that subnetworks collapse to nearly identical behavior. Increasing $k$ substantially increases prediction diversity, as expected. Regarding the EMA strength $\lambda$, the highest diversity consistently appears at $\lambda = 0.1$, and diversity decreases monotonically as $\lambda$ increases. This aligns with intuition: a larger $\lambda$ places stronger emphasis on previous EMA weights, suppressing variation across subnetworks and thereby reducing predictive diversity.

**Parameters budget.** Figure 5 compares the effect of the learnable parameter ratio—controlled by rank in LoRA and by mask sparsity in Random Mask, Mixout, and GMixout—on the ID–OOD trade-off. We evaluate ratios from 1% to 10% of learnable parameters. As the ratio increases, all methods trade off OOD for ID, with LoRA being the least sensitive. Across both domains, GMixout consistently yields higher OOD accuracy under any parameter budget while maintaining competitive ID accuracy on Real. On Sketch, it requires lower sparsity to match Random Mask and Mixout. Overall, optimizing about 10% of parameters is a good heuristic for balancing ID and OOD performance.

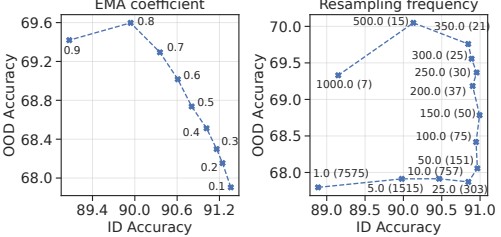

Figure 4: OOD-ID trade-off given variable $\lambda$ (left) and $k$ (right) on DomainNet Real.

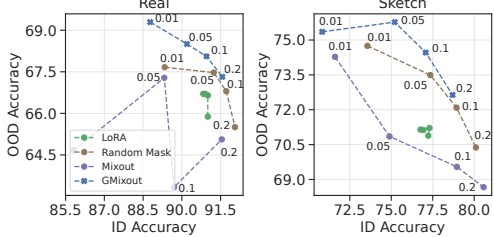

Figure 5: OOD-ID trade-off given the variable trainable parameters budget for PEFT methods.

Table 5: OOD accuracy on DomainNet Real and Sketch, showing the effect of pretrained weight source (left) and architecture size (right) on finetuning.

| Method | ViT-B/16 (IN-21k weights) | | | | ViT-L/14@336 (CLIP weights) | | | |
| | Real | | Sketch | | Real | | Sketch | |
| | ID | OOD | ID | OD | ID | OOD | ID | OD |
|---|---|---|---|---|---|---|---|---|
| Zero-shot | - | - | - | - | 89.5 | 74.2 | 74.6 | 79.2 |
| Full-FT | 91.5 | 55.1 | 78.6 | 58.0 | - | - | - | - |
| MovingAvg | 91.5 | 55.0 | 78.7 | 59.0 | - | - | - | - |
| Model Soups | 91.8 | 56.4 | 79.3 | 59.4 | - | - | - | - |
| Linear Probing | 90.1 | 49.0 | 69.8 | 57.8 | 93.3 | 71.0 | 81.9 | 73.3 |
| LoRA | 91.3 | 54.9 | 76.8 | 60.0 | **93.6** | 74.7 | **84.3** | 75.7 |
| Random Mask | 91.3 | 54.2 | 76.2 | 59.2 | 93.5 | 73.6 | 84.2 | 76.2 |
| Mixout | 91.6 | 55.8 | 77.8 | 59.4 | - | - | - | - |
| **GMixout (Ours)** | 90.4 | **65.0** | 77.0 | **60.3** | 93.4 | **76.2** | 82.1 | **81.4** |

**Vision only models.** To test GMixout beyond vision–language models, we evaluate it with ViT-B/16 pretrained on ImageNet-21k, initializing the classification head with linear probing weights (Kumar et al., 2022). As shown in Table 5(right), ImageNet-21k backbones yield lower OOD robustness than CLIP, but trends remain consistent: GMixout achieves the best balance among PEFT methods, with **65.0** OOD on Real and **60.3** on Sketch, outperforming LoRA, Mask, and Mixout while remaining competitive in ID accuracy.

**Large base models.** A key promise of PEFT is enabling finetuning of large models on consumer GPUs. To demonstrate efficiency, we finetune ViT-L/14 (428M parameters, 336 resolution) on a single RTX 3090 (24 GB VRAM), where full finetuning is infeasible and even PEFT methods exceed memory without strong parameter reduction. We therefore set LoRA rank to 4 and use sparsity 0.005 for Random Masking and GMixout (0.1% of total parameters). As shown in Table 5(left), ViT-L/14 achieves much higher ID and OOD accuracy than ViT-B/16, and GMixout again provides the best balance, reaching **76.2** OOD on Real and **81.4** on Sketch, outperforming other PEFT baselines while staying competitive on ID.

**Language Understanding.** For language understanding, we evaluate on CivilComments (Duchene et al., 2023), a standard benchmark for subpopulation shift (Koh et al., 2021; Yang et al., 2023). Each comment has a binary toxicity label and an 8-dimensional binary vector indicating references to eight demographic identities. This setting reflects an imbalance-recognition problem: labels are skewed, and the demographic attribute driving distribution shift is unknown during training, causing models to overfit majority groups and underperform on minority ones. We follow the WILDS splits (Koh et al., 2021), with demographic labels hidden during training. Following Yang et al. (2023), we fine-tune BERT$_{BASE}$ and report worst-group accuracy which defined as the minimum accuracy across demographic subpopulations and directly measuring robustness to distributional imbalance. We apply PEFT layers to all Linear layers in the transformer. As shown in Table 6, GMixout achieves the highest worst-group accuracy (69.0%), outperforming all seven fine-tuning baselines.

Table 6: Worst-group accuracy on CivilComments.

| Method | Worst acc. |
|---|---|
| Full-FT | 63.6 |
| Model Soups | 62.5 |
| LoRA | 58.5 |
| Random Mask | 56.8 |
| Mixout | 67.6 |
| DoRA | 64.5 |
| **GMixout (Ours)** | **69.0** |

## 6  CONCLUSIONS

In this paper, we revisit Mixout through an ensemble learning perspective and identify anchor choice, resampling frequency, and mask sparsity as key drivers of robustness. Building on these insights, we introduced GMixout, which leverages an adaptive EMA anchor, a resampling-frequency hyperparameter, and an efficient sparse-kernel implementation. Results on benchmarks spanning covariate shift, corruption, and class imbalance indicate that GMixout consistently enhances OOD robustness while maintaining strong ID accuracy. It surpasses baselines such as LoRA, Random Masking, and Model Soups, with no additional inference cost and practical efficiency on consumer GPUs. By combining the efficiency of PEFT with the robustness of ensembles, GMixout offers a practical and scalable solution for finetuning foundation models in real-world settings.

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

## 7 REPRODUCIBILITY STATEMENT

The code is included in the supplementary material, and all hyperparameters required to replicate the experiments are provided in Appendix A.5.

## A APPENDIX

Appendices include: a full treatment of the bias–variance–covariance–locality decomposition (Section A.1); additional ablation studies (Section A.2); comparison with prompt-tuning baselines under few-shot settings (Section A.3); training benchmark details (Section A.4); implementation details for reproducibility (Section A.5); and complete results on DomainNet and subsampled ImageNet-1k (Section A.6).

### A.1 BIAS–VARIANCE–COVARIANCE–LOCALITY DECOMPOSITION

Consider $M$ finetuned models, each trained under an identically distributed (i.d.) learning procedure $l_{P_{\mathrm{id}}}$. The expected error of a model with weights $\Phi_{\mathrm{WA}} = \frac{1}{M} \sum_{m=1}^{M} \Phi_m$ with respect to the joint distribution $L_{P_{\mathrm{id}}}^{M} = \{l_{P_{\mathrm{id}}}^{(m)}\}_{m=1}^{M}$ was decomposed in Rame et al. (2023) into:

$$\mathbb{E}_{L_{P_{\mathrm{id}}}^M}\big[\mathcal{L}_{\mathrm{ood}}(\Phi_{WA})\big] = \mathbb{E}_{(x,y)\sim P_{\mathrm{ood}}}\Big[\mathrm{bias}^2(x,y) + \frac{1}{M}\mathrm{var}(x) + \frac{M-1}{M}\mathrm{cov}(x)\Big] + O(\bar{\Pi}^2),$$

$$where\ \mathrm{bias}(x,y) = y - \bar{f}(x),$$

$$and\ \mathrm{var}(x) = \mathbb{E}_{l_{P_{\mathrm{id}}}^i}\Big[\big(f(x,\Phi_i) - \bar{f}(x)\big)^2\Big], \qquad (8)$$

$$and\ \mathrm{cov}(x) = \mathbb{E}_{l_{P_{\mathrm{id}}}^i, l_{P_{\mathrm{id}}}^j}\Big[\big(f(x,\Phi_i) - \bar{f}(x)\big)\big(f(x,\Phi_j) - \bar{f}(x)\big)\Big],$$

$$and\ \bar{\Pi}^2 = \mathbb{E}_{L_{P_{\mathrm{id}}}^M}\big[\max_i \|\Phi_i - \Phi_{WA}\|_2^2\big],$$

where $\bar{f}(x) = \mathbb{E}[f(x,\Phi_i)]$ denotes the expected prediction of $f$, $\mathrm{bias}(x,y)$ and $\mathrm{var}(x)$ are the bias and variance of the models with respect to a sample $(x,y)$, and $\mathrm{cov}(x)$ is the prediction covariance between two ensemble members whose weights are averaged. The locality term $\bar{\Pi}^2$ represents the expected squared maximum distance between individual weights and their weight average.

### A.2 ADDITIONAL ABLATIONS

**Measure forgetting.** To study how finetuning alters the backbone's representations, we evaluate transfer performance on five classification tasks: Flowers102 (Nilsback & Zisserman, 2008), Food101 (Bossard et al., 2014), Oxford Pets (Parkhi et al., 2012), DTD (Cimpoi et al., 2014), and Caltech101 (Li et al., 2007). In this setup, the finetuned backbone is frozen and paired with a new text-encoder-based classification head. Figure 6 shows that LoRA best preserves prior knowledge, followed by GMixout—an important property, as the goal is to adapt to new tasks without forgetting what has already been learned.

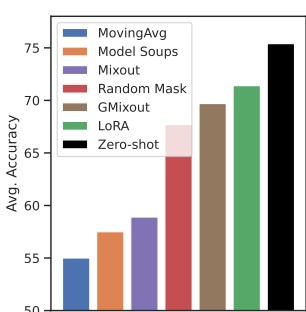

Figure 6: Average transfer accuracy on five out-of-task datasets after finetuning the methods on ImageNet-1k.

**Data Augmentation.** RandAugment (Cubuk et al., 2020) is a widely used data augmentation technique for improving robustness under distribution shift. It applies random perturbations to inputs using transformations such as cropping, color jittering, and geometric distortions. Table 7 reports the effect of applying RandAugment during finetuning, showing that data augmentation improves robustness across all baselines. For GMixout, the gains are particularly pronounced when training on the Sketch domain, yielding a 0.6-point improvement over the standard pipeline. This is intuitive, as Sketch data shares greater similarity with other rendition domains such as Painting and Clipart.

**Additional baselines.** In the main paper, we selected baselines that are widely recognized for delivering state-of-the-art performance under specific types of distribution shift. In particular, for corruption and covariate shift benchmarks, especially when the target distribution deviates substantially from the pretraining data, or when the dataset becomes large, Model Soups (Wortsman et al., 2022a) is a strong baseline (Rame et al., 2022; Fort et al., 2019). Many prior works Jang et al. (2025); Rame et al. (2023) explicitly acknowledge this and aim to match Model Soups' robustness while reducing computational cost. In this section, for broader comparisons, we additionally evaluate: **(1)** DoRA (Liu et al., 2024), which is a newer variant of LoRA, **(2)** LP-FT (Kumar et al., 2022) and AdamSPD (Tian et al., 2024), which are general robust fine-tuning methods, and **(3)** CLIPOOD (Shu et al., 2023), CaRot Oh et al. (2024), FLYP Goyal et al. (2023), which are robust fine-tuning methods specifically designed for vision-language models.

We evaluate all methods on two DomainNet settings (trained on Real or Sketch and evaluated on the remaining domains) as well as on iWildCam. Furthermore, we run GMixout with three different random seeds to demonstrate its robustness to stochastic variation. Table 8 summarizes these extended results alongside the performance of our main baselines.

Table 7: OOD accuracy on DomainNet Real and Sketch with applying RandAugment during fine-tuning.

| Method | ID | OOD | | | | ID | OOD | | | |
|---|---|---|---|---|---|---|---|---|---|---|
| | Real | Clipart | Sketch | Painting | Avg. | Sketch | Real | Clipart | Painting | Avg. |
| Zero-shot | 86.1 | 69.3 | 62.0 | 66.1 | 65.8 | 65.4 | 82.9 | 69.3 | 66.1 | 72.8 |
| **+RandAugment** | | | | | | | | | | |
| Full-FT | 91.7 | 68.0 | 55.0 | 61.2 | 61.4 | 80.7 | 65.9 | 68.3 | 54.5 | 63.0 |
| MovingAvg | 91.7 | 67.9 | 55.0 | 61.2 | 61.4 | 80.7 | 66.3 | 68.4 | 54.7 | 63.1 |
| Model Soups | 92.2 | 70.8 | 59.6 | 64.7 | 65.0 | **81.7** | 70.6 | 71.1 | 59.5 | 67.0 |
| Linear Probing | 90.4 | 66.8 | 59.0 | 63.6 | 63.1 | 74.2 | 72.5 | 64.8 | 57.7 | 65.0 |
| LoRA | 90.6 | 72.0 | 63.1 | 66.3 | 67.1 | 76.0 | 79.0 | 71.4 | 65.7 | 72.0 |
| Random Mask | 91.5 | 72.3 | 63.1 | 66.3 | 67.3 | 78.0 | 79.6 | 72.5 | 66.0 | 72.7 |
| Mixout | 92.0 | 69.7 | 57.9 | 63.3 | 63.6 | 78.9 | 76.5 | 71.9 | 63.3 | 70.6 |
| **GMixout (Ours)** | 90.9 | 72.6 | 63.5 | 66.8 | **67.7** | 76.2 | 82.4 | 73.5 | 69.3 | **75.0** |

Table 8: IID-OOD accuracy on DomainNet (Real and Sketch) and micro-F1 on iWildCam with additional baselines.

| Method | DomainNet | | | | iWildCam | |
|---|---|---|---|---|---|---|
| | Real | | Sketch | | | |
| | ID | OOD | ID | OD | ID | OOD |
| Zero-shot | 86.1 | 65.8 | 65.4 | 72.8 | 11.5 | 12.7 |
| Full-FT | 91.6 | 60.2 | 80.7 | 62.4 | 46.5 | 36.4 |
| MovingAvg | 91.6 | 60.2 | 80.7 | 62.3 | 46.6 | 34.2 |
| Model Soups | 92.4 | 64.3 | 81.9 | 65.9 | 46.8 | **37.6** |
| Linear Probing | 90.7 | 62.2 | 75.7 | 64.4 | 31.6 | 24.7 |
| LoRA | 90.9 | 66.5 | 76.9 | 71.0 | 49.7 | 35.9 |
| Random Mask | 91.7 | 66.8 | 78.9 | 72.0 | 47.9 | 35.2 |
| Mixout | 89.7 | 66.5 | 79.1 | 69.8 | 43.5 | 32.9 |
| LP-FT | 91.6 | 59.3 | 80.7 | 61.2 | 47.4 | 31.8 |
| DoRA | 91.1 | 66.0 | 77.7 | 70.6 | 50.4 | 35.7 |
| AdamSPD | 92.5 | 66.2 | 81.6 | 68.5 | 47.4 | 36.1 |
| CLIPOOD | 91.7 | 60.2 | 80.4 | 62.6 | 41.6 | 24.7 |
| CaRot | 91.6 | 67.8 | 76.3 | 73.7 | - | - |
| FLYP | 90.0 | 57.9 | 78.7 | 63.9 | - | - |
| **GMixout (Ours)** | 90.9$_{\pm0.08}$ | **68.4**$_{\pm0.19}$ | 76.8$_{\pm0.25}$ | **75.6**$_{\pm0.31}$ | 49.9$_{\pm0.39}$ | 37.7$_{\pm0.73}$ |

The results show, none of the newly added baselines surpass GMixout. The only method that comes close is CaRot, which performs competitively but requires full-parameter training of both the vision and language towers, leading to substantially higher computational and memory costs. In contrast, GMixout delivers strong robustness while activating only 10% of the parameters, offering a far more efficient trade-off.

**WISE-FT extension.** An effective way to combine the downstream performance of finetuned models with the robustness of zero-shot models is to ensemble their weights (Wortsman et al., 2022b). This technique, known as WISE-FT, can be applied to any finetuning method. By adjusting a merging coefficient, WISE-FT trades ID accuracy for OOD robustness. Table 9 reports results on DomainNet (Real and Sketch) and iWildCam with a coefficient of 0.5. On DomainNet, where the zero-shot model is already strong, WISE-FT improves all baselines, especially on OOD. However, on iWildCam, where CLIP performs poorly in zero-shot, WISE-FT provides no benefit—in fact, the baselines perform better without it. In contrast, GMixout consistently improves OOD performance regardless of the zero-shot model's strength.

**Extended training on ImageNet-1k.** An easy way to improve downstream performance is to train for longer. However, this often reduces robustness, particularly when starting from a strong

Table 9: OOD accuracy on DomainNet (Real and Sketch) and micro-F1 on iWildCam with WISE-FT with merging coefficient 0.5.

| Method | DomainNet | | | | iWildCam | |
| | Real | | Sketch | | | |
| | ID | OOD | ID | OD | ID | OOD |
|---|---|---|---|---|---|---|
| Zero-shot | 86.1 | 65.8 | 69.3 | 72.8 | 11.5 | 12.7 |
| Full-FT | 92.0 | 68.9 | 80.4 | 75.2 | 38.6 | 31.1 |
| MovingAvg | 92.0 | 68.9 | 80.5 | 75.2 | 38.7 | 31.2 |
| Model Soups | **92.2** | 69.9 | **80.7** | 75.7 | 40.1 | 32.3 |
| LoRA | 89.0 | 69.0 | 73.1 | 75.6 | 22.4 | 23.2 |
| Random Mask | 90.6 | 69.7 | 76.3 | 76.1 | 42.2 | 32.7 |
| Mixout | 89.5 | 69.3 | 78.2 | 75.7 | 38.8 | 31.0 |
| **GMixout (Ours)** | 90.1 | **69.9** | 76.0 | **76.4** | **43.1** | **34.2** |

Table 10: Accuracy on ImageNet-1k (ID) and four natural distribution shifts (OOD) under extended finetuning, along with transfer performance on five out-of-task datasets.

| Method | ID | OOD | | | | | Out-of-Task |
| | IN-1k | IN-V2 | IN-R | IN-Sketch | IN-A | Avg. | Avg. of 5 Datasets |
|---|---|---|---|---|---|---|---|
| Zero-shot | 68.2 | 62.0 | 76.4 | 46.6 | 51.6 | 59.1 | 75.4 |
| **10 Epochs** | | | | | | | |
| LoRA | 83.7 | 74.0 | 69.3 | 49.6 | 46.6 | 59.9 | 71.4 |
| Random Mask | 83.2 | 74.0 | 70.4 | 49.9 | 47.7 | 60.5 | 67.7 |
| Mixout | 82.7 | 72.6 | 62.7 | 45.0 | 37.4 | 54.4 | 58.9 |
| **GMixout (Ours)** | 82.8 | 73.6 | 71.6 | 50.4 | 47.2 | 60.7 | 69.7 |
| **30 Epochs** | | | | | | | |
| LoRA | 83.5 | 74.0 | 68.9 | 49.6 | 47.2 | 59.9 | 70.2 |
| Random Mask | 83.2 | 73.8 | 69.1 | 49.2 | 47.3 | 59.9 | 66.4 |
| Mixout | 81.8 | 71.2 | 58.4 | 41.3 | 32.4 | 50.8 | 55.5 |
| **GMixout (Ours)** | 83.1 | 73.8 | 70.4 | 50.1 | 46.1 | 60.1 | 68.0 |

foundation model. In Table 10, we extend training on ImageNet-1k for 30 epochs. As shown, LoRA and Random Mask show no ID improvement, suggesting overfitting, while GMixout continues to improve ID accuracy. OOD accuracy drops across all methods, but GMixout is the most resilient. A similar trend holds for out-of-task datasets, where all methods experience decreased performance.

## A.3 COMPARISON WITH PROMPT-TUNING

In this section, we compare GMixout with prompt-tuning baselines, including CoOp (Zhou et al., 2022b) and CoCoOp (Zhou et al., 2022a), on the few-shot learning benchmark. CoOp replaces hand-crafted prompts in vision–language models, such as CLIP, with learnable context vectors trained end-to-end, thereby improving performance but showing limited generalization to unseen classes. CoCoOp extends this by employing a meta-network to generate image-conditioned context tokens, enhancing generalization at the cost of slower inference. Following the evaluation protocol of Zhou et al. (2022a), we sample 16 shots per class from ImageNet-1k for training and evaluate on both the ID validation set and four natural distribution shifts derived from ImageNet-1k. We use ViT-B/16 with CLIP weights as the image encoder. We use the mask sparsity of $0.01$ for GMixout. As shown in Table 11, GMixout consistently outperforms prompt-tuning baselines on both ID and OOD, except on IN-A.

## A.4 BENCHMARK DETAILS

For benchmarking, we use PyTorch v2.7.1 profiling tools (Paszke et al., 2019). All experiments are run on a ViT-B/16 backbone with an NVIDIA RTX 3090 (24 GB VRAM) and batch size 64. We set LoRA rank to $r = 64$, and for Random Masking, Mixout, and GMixout, we use mask sparsity $s = 0.1$, optimizing about 10% of ViT-B/16 total parameters (85M).

Table 11: Accuracy on few-shot ImageNet-1k (ID, 16 shots per class) and four natural distribution shifts (OOD).

| Method | ID | OOD | | | | |
| --- | --- | --- | --- | --- | --- | --- |
| | IN-1k | IN-V2 | IN-R | IN-Sketch | IN-A | Avg. |
| Zero-shot | 68.2 | 62.0 | 76.4 | 46.6 | 51.6 | 59.1 |
| CoOp (Zhou et al., 2022b) | 71.5 | 64.2 | 75.2 | 48.0 | 49.7 | 59.3 |
| CoCoOp (Zhou et al., 2022a) | 71.0 | 64.0 | 76.2 | 48.7 | 50.6 | 59.9 |
| **GMixout (Ours)** | **73.4** | **65.8** | **77.3** | **49.0** | 49.3 | **60.3** |

## A.5 IMPLEMENTATION DETAILS

We train all methods with the AdamW optimizer (Loshchilov & Hutter, 2019), using weight decay $0.1$. Batch size is $128$ for all datasets except ImageNet-1k, where it is $512$. To initialize the linear classifier in CLIP-based ImageNet-1k experiments, we use the prompt templates provided in Radford et al. (2021). For all other datasets, we adopt the template "a photo of a `<class>`" to obtain the corresponding embeddings from the CLIP text encoder. For each baseline, we tune learning rates from $\{1e-2, 1e-3, 1e-4, 1-5\}$ with a cosine decay scheduler that anneals to $0$ after a 1-epoch warmup. Training runs for 10 epochs on ImageNet-1k and DomainNet, and 20 epochs on all other datasets. The best checkpoint is selected using each dataset's in-domain validation set (Gulrajani & Lopez-Paz, 2021).

For long-tail benchmarks, we adopt the *Logit Adjustment* (LA) loss (Menon et al., 2021), which mitigates head-class bias by adding a label-dependent offset to the logits:

$$\ell_{LA}(y, f(\boldsymbol{x})) = -\log \frac{\exp(g_y(\boldsymbol{x}) + \log \pi_y)}{\sum_{y' \in \mathcal{C}} \exp(f_{y'}(\boldsymbol{x}) + \log \pi_{y'})}, \tag{9}$$

where $\pi \in \Delta_y$ are estimates of the class priors $P(y)$ based on the empirical class frequencies on the training data $D$. In a recent study, Shi et al. (2024) observed that starting from CLIP pretrained weights, the LA loss alone is insufficient to achieve strong performance.

## A.6 DETAIL RESULTS

In this section, we show detailed results of our main experiments. Tables 12 and 13 report the full results on the Real, Sketch, Clipart, and Painting domains of DomainNet, treating each as ID with their corresponding OOD domains. Table 14 provides the detailed ID–OOD results corresponding to Figure 2, using the ImageNet-1k subsampled training sets with 25, 50, 100, and 200 images per class, respectively.

Table 12: OOD accuracy on DomainNet Real and Sketch.

| Method | ID | OOD | | | | ID | OOD | | | |
| --- | --- | --- | --- | --- | --- | --- | --- | --- | --- | --- |
| | Real | Clipart | Sketch | Painting | Avg. | Sketch | Real | Clipart | Painting | Avg. |
| Zero-shot | 86.1 | 69.3 | 62.0 | 66.1 | 65.8 | 65.4 | 82.9 | 69.3 | 66.1 | 72.8 |
| Full-FT | 91.6 | 67.0 | 53.3 | 60.2 | 60.2 | 80.7 | 65.7 | 67.7 | 53.7 | 62.4 |
| MovingAvg | 91.6 | 67.0 | 53.4 | 60.2 | 60.2 | 80.7 | 65.7 | 67.8 | 53.5 | 62.3 |
| Model Soups | 92.4 | 70.5 | 58.2 | 64.1 | 64.3 | 81.9 | 69.0 | 70.7 | 58.0 | 65.9 |
| Linear Probing | 90.7 | 66.0 | 57.9 | 62.7 | 62.2 | 75.7 | 71.4 | 65.4 | 56.4 | 64.4 |
| LoRA | 90.9 | 71.7 | 62.3 | 65.6 | 66.5 | 76.9 | 77.8 | 71.5 | 63.6 | 71.0 |
| Random Mask | 91.7 | 72.1 | 62.2 | 66.1 | 66.8 | 78.9 | 78.8 | 72.9 | 64.6 | 72.0 |
| Mixout | 89.7 | 71.5 | 62.4 | 65.7 | 66.5 | 79.1 | 75.2 | 71.3 | 62.8 | 69.8 |
| **GMixout (Ours)** | 90.8 | 73.6 | 64.6 | 67.9 | **68.7** | 77.0 | 82.6 | 74.1 | 69.3 | **75.3** |

## A.7 PREDICTION DIVERSITY ANALYSIS

Diversity among ensemble members is widely recognized as a key driver of ensemble effectiveness (Dietterich, 2000; Rame et al., 2022; Fort et al., 2019; Izmailov et al., 2018). In Figure 7, we

Table 13: OOD accuracy on DomainNet Painting and Clipart.

| Method | ID Painting | OOD Clipart | Sketch | Real | Avg. | ID Clipart | OOD Sketch | Painting | Real | Avg. |
|---|---|---|---|---|---|---|---|---|---|---|
| Zero-shot | 69.3 | 69.3 | 62.0 | 82.9 | 71.4 | 69.5 | 62.0 | 66.1 | 82.9 | 70.4 |
| Full-FT | 83.6 | 62.1 | 51.2 | 71.6 | 61.6 | 85.0 | 57.9 | 55.0 | 72.7 | 61.9 |
| MovingAvg | 83.6 | 63.0 | 51.5 | 71.7 | 62.0 | 84.9 | 58.0 | 54.9 | 72.5 | 61.8 |
| Model Soups | 84.7 | 65.2 | 54.9 | 73.6 | 64.6 | 86.3 | 61.6 | 58.7 | 75.1 | 65.2 |
| Linear Probing | 80.6 | 58.4 | 53.2 | 68.9 | 60.2 | 81.2 | 57.8 | 56.8 | 73.6 | 62.7 |
| LoRA | 81.5 | 68.1 | 61.4 | 77.6 | 69.0 | 81.8 | 64.0 | 64.1 | 80.4 | 69.5 |
| Random Mask | 82.8 | 68.4 | 59.9 | 78.3 | 68.9 | 83.2 | 64.5 | 64.5 | 80.4 | 69.8 |
| Mixout | 82.5 | 65.1 | 55.8 | 74.3 | 65.0 | 84.9 | 62.3 | 60.7 | 77.6 | 66.9 |
| **GMixout (Ours)** | 81.3 | 71.9 | 66.0 | 82.8 | **73.5** | 81.8 | 67.6 | 69.0 | 83.7 | **73.4** |

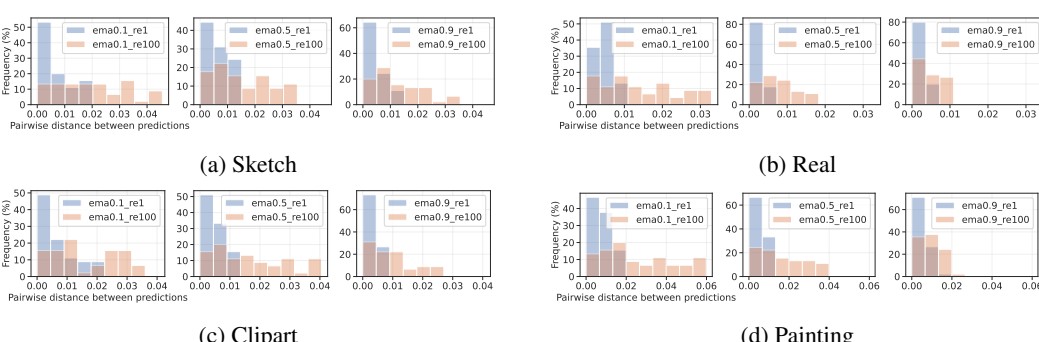

(a) Sketch  (b) Real

(c) Clipart  (d) Painting

Figure 7: Frequencies of prediction diversities (Aksela, 2003) between two subnetworks obtained from a single training run on DomainNet-Sketch, under $\lambda \in \{0.1, 0.5, 0.9\}$ and $k \in \{1, 100\}$. Across both in-domain (a) and out-of-domain (b, c, d) evaluations, setting the resampling parameter to $k = 1$ yields virtually no diversity for many of the checkpoint pairs. Increasing $k$ substantially boosts prediction diversity, as expected. With respect to the GMixout EMA strength $\lambda$, the highest diversity occurs at $\lambda = 0.1$, and diversity decreases monotonically as $\lambda$ increases.

validate that GMixout also benefits from such diversity. We quantify prediction diversity using the ratio-error metric (Aksela, 2003), defined as the ratio $N_{\text{diff}}/N_{\text{simul}}$ between the number of different errors $N_{\text{diff}}$ and simultaneous errors $N_{\text{simul}}$ on test samples for each pair of checkpoints obtained during GMixout training on DomainNet-Sketch, under $\lambda \in \{0.1, 0.5, 0.9\}$ and $k \in \{1, 100\}$. A higher average over the $\binom{M}{2}$ checkpoint pairs indicates that members are less likely to make the same errors.

For $k = 100$, we extract the weights from the last ten re-masking steps of GMixout. To ensure fair comparison, we extract an equivalent number of checkpoints for the $k = 1$ setting over the same training interval. Across both in-domain evaluations (Figure 7a) and out-of-domain evaluations (Figures 7b, 7c, and 7d), the trend is consistent: when $k = 1$, prediction diversity is essentially zero for many of the checkpoint pairs, indicating that subnetworks collapse to nearly identical behavior. Increasing $k$ substantially increases prediction diversity, as expected. Regarding the EMA strength $\lambda$, the highest diversity consistently appears at $\lambda = 0.1$, and diversity decreases monotonically as $\lambda$ increases. This aligns with intuition: a larger $\lambda$ places stronger emphasis on previous EMA weights, suppressing variation across subnetworks and thereby reducing predictive diversity.

Table 14: ID and OOD results on ImageNet-1k subsampled training datasets (25, 50, 100, and 200)-shots.

| Method | ID | OOD | | | | |
|---|---|---|---|---|---|---|
| | IN-1k | IN-V2 | IN-R | IN-Sketch | IN-A | Avg. |
| Zero-shot | 68.2 | 62.0 | 76.4 | 46.6 | 51.6 | 59.1 |
| **25 Shots** | | | | | | |
| Full-FT | 72.7 | 63.4 | 59.3 | 40.2 | 29.4 | 48.1 |
| MovingAvg | 72.9 | 63.5 | 61.6 | 41.1 | 29.6 | 48.9 |
| Model Soups | 74.5 | 65.8 | 62.7 | 43.5 | 35.0 | 51.8 |
| Linear Probing | 72.0 | 63.2 | 72.4 | 43.8 | 47.6 | 56.7 |
| LoRA | 75.1 | 66.8 | 72.8 | 46.5 | 44.1 | 57.6 |
| Random Mask | 73.9 | 66.3 | 71.4 | 47.2 | 44.4 | 57.3 |
| Mixout | 74.2 | 66.3 | 66.2 | 44.7 | 39.7 | 54.2 |
| **GMixout (Ours)** | 73.5 | 65.9 | 74.6 | 48.6 | 44.6 | 58.4 |
| **50 Shots** | | | | | | |
| Full-FT | 74.7 | 65.3 | 57.0 | 39.7 | 28.6 | 47.7 |
| MovingAvg | 74.7 | 63.4 | 76.1 | 47.1 | 51.6 | 59.6 |
| Model Soups | 76.5 | 67.4 | 62.1 | 44.0 | 34.4 | 52.0 |
| Linear Probing | 73.3 | 63.8 | 71.2 | 43.3 | 46.5 | 56.2 |
| LoRA | 76.8 | 68.3 | 71.6 | 46.0 | 43.2 | 57.3 |
| Random Mask | 75.8 | 67.9 | 70.4 | 47.6 | 44.2 | 57.5 |
| Mixout | 76.0 | 67.2 | 65.1 | 44.8 | 39.3 | 54.1 |
| **GMixout (Ours)** | 75.2 | 67.3 | 72.8 | 48.3 | 44.9 | 58.4 |
| **100 Shots** | | | | | | |
| Full-FT | 76.9 | 66.7 | 58.2 | 41.3 | 28.9 | 48.8 |
| MovingAvg | 76.9 | 63.3 | 76.0 | 46.8 | 51.2 | 59.3 |
| Model Soups | 78.4 | 69.0 | 62.8 | 45.5 | 35.3 | 53.1 |
| Linear Probing | 74.7 | 64.6 | 70.3 | 43.2 | 45.6 | 55.9 |
| LoRA | 78.5 | 69.0 | 70.9 | 46.2 | 44.0 | 57.6 |
| Random Mask | 77.6 | 69.0 | 70.2 | 47.9 | 43.9 | 57.7 |
| Mixout | 77.6 | 68.5 | 65.3 | 45.4 | 38.0 | 54.3 |
| **GMixout (Ours)** | 76.9 | 68.9 | 71.9 | 48.1 | 44.5 | 58.4 |
| **200 Shots** | | | | | | |
| Full-FT | 78.7 | 68.6 | 57.9 | 41.2 | 29.5 | 49.3 |
| MovingAvg | 78.7 | 63.4 | 75.5 | 46.2 | 50.8 | 59.0 |
| Model Soups | 80.2 | 70.5 | 63.3 | 46.0 | 35.7 | 53.9 |
| Linear Probing | 75.8 | 65.6 | 70.0 | 43.4 | 45.9 | 56.2 |
| LoRA | 79.8 | 70.2 | 70.7 | 46.8 | 45.1 | 58.2 |
| Random Mask | 79.4 | 70.6 | 70.5 | 47.9 | 45.3 | 58.6 |
| Mixout | 79.1 | 69.9 | 64.4 | 45.0 | 36.9 | 54.0 |
| **GMixout (Ours)** | 78.5 | 70.2 | 71.3 | 48.0 | 45.0 | 58.7 |

