# OpenReview forum: "Revisiting Mixout: An Overlooked Path to Robust Finetuning"
_ICLR.cc/2026/Conference — Submitted to ICLR 2026_

### Official Review · Reviewer_26yR · 2025-10-24

**Soundness:** 2
**Presentation:** 3
**Contribution:** 1
**Rating:** 2
**Confidence:** 5

**Summary:**

This paper revisits Mixout (Lee et al.) for robust fine-tuning of a pre-trained vision model under distribution shifts between train and test data. The authors motivate themselves by first reminding the implicit L2-penalty behavior of Mixout and then proposing a better alternative to Mixout, GMixout, by grounding it with an expected OOD loss decomposition proposed in the DiWA paper (Rame et al. 2022). By introducing an exponential moving average anchor with dynamic masking per episode, the proposed method enjoys a better tradeoff between ID and OOD performance compared to the considered baseline.


---

> Reference
- Lee at al. 2023, "Mixout: Effective Regularization to Finetune Large-scale Pretrained Language Models"
- Rame et al. 2022, "Diverse Weight Averaging for Out-of-Distribution Generalization"

**Strengths:**

- Revisiting Mixout (Lee et al. 2023) in the robust fine-tuning setup offers interesting insights to the community.
- The proposed method is naturally connected to Mixout and DiWA (Rame et al. 2022), which have nice theoretical properties.
- The proposed method shows a performance gain across multiple distribution shift scenarios.
- The paper writing is clear and well-organized.

---

> Reference
- Lee et al. 2023, "Mixout: Effective Regularization to Finetune Large-scale Pretrained Language Models"
- Rame et al. 2022, "Diverse Weight Averaging for Out-of-Distribution Generalization"

**Weaknesses:**

- `Lack of technical/theoretical innovation`
  - The core technical contribution of the proposed method is the episodic update of the anchor (which is fixed as the pre-trained model weight in the original Mixout (Lee et al. 2023) and the Mask)
  - However, the idea of moving the anchor is already well explored by previous works on robust fine-tuning (Jang et al. 2024 -- periodic merging), and the merits of the exponential moving average (EMA) during fine-tuning are also well-explored by existing methods (Shu et al. 2023, Oh et al. 2024)
  - Although the authors bring a theory from DiWA paper (Rame et al. 2022) to explain the desired property of GMixout, there is **no rigorous analysis** (including supplementary A.1 that only provides a detailed decomposition) of **why GMixout achieves good control on the combination of variance-covariance-locality terms.**
- `Too limited baseline lineup and less comprehensive literature review`
  - In the experiment design, the authors **do not include some very representative robust fine-tuning methods**, such as WiSE-FT (Wortsman et al. 2022), LP-FT (Kumar et al. 2022), and FLYP (Goyal et al. 2022), which makes it difficult to gauge how significant the performance gain achieved by GMixout is.
  - Besides, they do not even mention some relevant works, CLIPood (Shu et al. 2023) and CaRot (Oh et al. 2024), where the **exponential moving average (EMA) style parameter update** was leveraged for the robust fine-tuning context, and VRF (Zhu et al. 2024), which **reduces variance for OOD robustness via ensemble between pre-trained and fine-tuned model prediction**, and DaWin (Oh et al. 2025), a state-of-the-art robust-fine-tuning method that is based on the **weight interpolation between pre-trained and fine-tuned models**.
    - Lack of citing these highly relevant works raises concern about the completeness of the authors' literature review.
- `(minor) Incorrect learnable parameter description`
  - I think the proposed method should have the same learnable parameters as Mixout (85.5 M) in Table 3.
  - Although the per-step updated parameter can be 9M (only survives after masking), as the authors sample the new mask for every episode, each episode has different updated parameters.
  - Therefore, the learnable parameter over the whole training should be counted as 85.5 M (the same as the full FT).
  - And the GMixout is hard to recognize as a PEFT method, thereby.
---

> Reference
- Rame et al. 2022, "Diverse Weight Averaging for Out-of-Distribution Generalization"
- Jang et al. 2024, "Model Stock: All we need is just a few fine-tuned models"
- Wortsman et al. 2022, "Robust fine-tuning of zero-shot models"
- Kumar et al. 2022, "Fine-Tuning can Distort Pretrained Features and Underperform Out-of-Distribution"
- Goyal et al. 2022, "Finetune like you pretrain: Improved finetuning of zero-shot vision models"
- Shu et al. 2023, "CLIPood: Generalizing CLIP to Out-of-Distributions"
- Oh et al. 2024, "Towards Calibrated Robust Fine-Tuning of Vision-Language Models"
- Zhu et al. 2024, "Robust Fine-tuning of Zero-shot Models via Variance Reduction"
- Oh et al. 2025, "DaWin: Training-free Dynamic Weight Interpolation for Robust Adaptation"

**Questions:**

Please see the weakness section, and feel free to refute if there is any misunderstanding from me.

---

> ### Author Response · Authors · 2025-11-21
> **Response to Reviewer 26yR**
>
> We thank the reviewer for their thoughtful feedback and for highlighting the clarity of our presentation, the theoretical connection to Mixout and DiWA, and the consistent gains under distribution shift. We address their questions below:
>
> > ### W1.1 Lack of technical/theoretical innovation
>
> Please refer to the ***Lack of innovation*** section in the **Global response [1/2]**.
>
> > ### W1.2 why GMixout achieves good control on the combination of variance-covariance-locality terms.
>
> Section 4.1 develops our intuition using the BVCL perspective. In summary, increasing the resampling frequency controls how often subnetworks are refreshed, and, together with mask sparsity, reduces subnetwork correlation, allowing GMixout to recover genuine ensemble benefits. Figures 1, 3, and 4 empirically support this intuition. To further examine GMixout’s implicit ensembling behavior, we also added a new analysis in Appendix A.7 (highlighted in blue) measuring prediction diversity across its subnetworks.
>
> We measure subnetwork diversity using the ratio-error metric, computed across all checkpoint pairs from GMixout runs on DomainNet-Sketch with $\lambda \in \{0.1,0.5,0.9\}$ and $k \in \{1,100\}$. Higher values mean subnetworks tend not to make the same mistakes. We find that:
>
> - with $k=1$, prediction diversity is virtually zero for most checkpoint pairs; the subnetworks behave almost identically;
> - increasing $k$ markedly boosts diversity;
> - for fixed $k$, diversity is maximized at $\lambda=0.1$ and decreases as $\lambda$ increases.
>
> Overall, GMixout enhances robustness by producing a more diverse set of hypotheses through frequent resampling.
>
>
> > ### W2. Too limited baseline lineup and less comprehensive literature review.
>
> Please refer to the ***More comprehensive baselines*** section in the **Global response [2/2]** for full details. Here, we briefly highlight the key points. We expand our baselines to six additional method:
>
> 1. PEFT method (**DoRA**),
> 2. General robust fine-tuning methods (**LP-FT** and **AdamSPD**), and
> 3. Recent robust fine-tuning methods specifically designed for vision-language models (**CLIPOOD**, **CaRot**, **FLYP**).
> 4. We also for reference, include **WiseFT**, a general model-averaging baseline previously detailed in Appendix A.2 (under WISE-FT extension).
>
> We evaluate all methods on two DomainNet settings (trained on Real or Sketch and evaluated on the remaining domains) as well as on iWildCam. As shown in the table in the global response, **none of the new baselines outperform GMixout**.
>
> Please note that we exclude DaWin [1] and VRF [2] for the following reasons:
> - Both methods rely on unlabeled test samples to optimize the merging coefficients in a WiseFT-style model combination. This differs from our setting, where we assume no access to test data during adaptation.
> - Both methods are extremely inefficient at inference time, as they compute a new coefficient for every individual sample, effectively producing a distinct set of model weights per input. This prevents batch processing and negates the efficiency benefits of GPU parallelism.
>
> For these reasons, DaWin and VRF are not directly comparable to our method.
>
> *[1] Oh, Changdae, et al. "Dawin: Training-free dynamic weight interpolation for robust adaptation." ICLR 2025.*
>
> *[2] Zhu, Beier, Jiequan Cui, and Hanwang Zhang. "Robust fine-tuning of zero-shot models via variance reduction." Neurips 2024.*
>
> > ### W3.1 Incorrect learnable parameter description
>
> We agree with the reviewer that the total number of learnable parameters in GMixout is identical to that of Mixout (85.5M), and we never claimed otherwise. However, as noted in L374, the parameter counts in Table 3 reflect the number of parameters updated per gradient step, not the total parameter count. This distinction is crucial, as it determines whether a method can run on consumer-grade GPUs.
>
> GMixout achieves this through its sparse-kernel implementation, in which only the unmasked parameters are updated. This reduction in per-step computation is what makes GMixout feasible on limited hardware.
>
> > ### W3.2 GMixout is hard to recognize as a PEFT method
>
> As noted in prior work [1], *“PEFT refers to the process of adjusting the parameters of a pre-trained large model to adapt it to a specific task or domain while minimizing the number of additional parameters introduced or computational resources required.”* Under this definition, GMixout naturally qualifies as a PEFT method: its sparse updates enable efficient training on consumer-grade GPUs. We further reinforce this through our empirical results on ViT-L/14 (428M parameters, 336-resolution) in Table 5. In particular, we fine-tune all PEFT baselines, including GMixout, on a single RTX 3090 (24 GB VRAM), where full fine-tuning is infeasible and even PEFT methods exceed memory limits without substantial parameter reduction.
>
> *[1] Han, Zeyu, et al. "Parameter-efficient fine-tuning for large models: A comprehensive survey." TMLR 2024.*

---

> ### Comment · Reviewer_26yR · 2025-11-28
>
> I really appreciate the authors' professional rebuttal! After carefully reviewing the newly added A.7 and comparing it with SOTA robust fine-tuning methods, my impression of this paper has shifted significantly towards the positive side.
> **I WANT TO RAISE MY RATING ACCORDINGLY, BUT THE `EDIT REVIEW` BUTTON HAS DISAPPEARED**. I hope the PC recognizes this issue soon and allows me to adjust my rating.
>
> But there are some remaining suggestions/comments below
>
> ---
>
> * Putting new results and explanation, revising the main text directly in the main body of the paper, rather than deferring them to the appendix or confining them to this rebuttal.
>   * First of all, I want the authors to check the `ICLR'26 authors' guidelines`. **It allows authors to add one additional page (up to 10) during the rebuttal period.** Why don't the authors directly enhance the main body of the paper?
>   * I think the authors' additional experiment that compares their GMixout to other representative robust finetuning methods is key to ensuring the persuasiveness of the paper. Besides, although the author put extra justification on their theoretical analysis in Appendix A.7, I think it should be presented in the main body of the paper, not the appendix.
>   * I am wondering why the authors do not provide this experimental result/setup and a clearer theoretical justification in their revised draft.
> * less comprehensive literature review.
>   * I appreciate the author's rebuttal about the reason why they do not compare their method with some ensemble methods, such as DaWin and VRF.
>   * Yeah, totally agree, I don't mean the comparison with these methods, but my point was that at least acknowledging these lines of research, trying to ensemble the knowledge of pre-training and fine-tuning model weights to address the robustness problem -- which is exactly connected to the motivation of your proposed method in the high-level goal; But if it seems too aggresive abstraction, feel free to neglect this comment!
> * Incorrect learnable parameter description
>   * Thanks for the clarification on 'per-gradient-step vs. entire', but I don't see why the per-gradient-step trainable parameter matters then (given that total trainable parameters across entire training is the same with Mixout)? The more important metrics may be the trainable parameters across all training and other speed/memory-related quantities.
>   * Could you further clarify why per-step learnable parameters matter? Have I missed some important facts?
>
> ---
>
> Anyway, if the authors revise the main body of the paper manuscript accordingly, I am willing to raise my score (the openreivew system does not allow it right now, though).

---

> ### Author Response · Authors · 2025-11-28
> **Response to Reviewer 26yR**
>
> Thank you for your thoughtful follow-up and the positive reassessment. We sincerely appreciate the time and care you put into reviewing our rebuttal and additional analyses. We address your comments below:
>
> > ### Revised manuscript
>
> Thank you for the suggestion, we have updated the manuscript accordingly. In particular:
> 1. We have incorporated the new prediction-diversity analysis and language-understanding experiments into the main body of the paper.
> 2. Due to the space constraints, we have added the additional baseline results to the Appendix. However, for the camera-ready version, we plan to extend these evaluations to the full DomainNet and CIFAR100-C datasets and update Table 1 accordingly.
> 3. We have updated the related-work section to reflect the new benchmarks and analyses.
>
> We have highlighted all new changes in blue. Please note that we are retaining a duplicate of Appendix A.7 in the revised version of the manuscript to maintain clarity for other reviewers. We will consolidate it in the camera-ready version.
>
> > ### Comprehensive literature review.
>
> We appreciate the reviewer’s suggestion and have updated our literature review accordingly.
>
> > ### Could you further clarify why per-step learnable parameters matter?
>
> Neural network optimization almost always operates on mini-batches, which makes the cost of a single forward–backward pass the primary bottleneck for fine-tuning large models on consumer-grade GPUs. Full-parameter fine-tuning requires storing gradients and activations for every parameter in the network. In contrast, PEFT methods reduce this cost in different ways. For example, with LoRA, gradients are computed only for the low-rank matrices and their associated activations; with masking-based approaches that leverage sparse kernels, only the unmasked parameters participate in backpropagation.
>
> Consequently, the total number of trainable parameters is less important than the number of parameters involved in each forward–backward step. In Table 3, we report the per-gradient-step parameter count rather than the total number of tunable parameters to highlight exactly this distinction.
>
> > ### Incorrect learnable parameter description.
>
> The original Mixout implementation does not use sparse kernels; it simply zeroes out gradients during the backward pass but still requires storing all gradients. In contrast, GMixout leverages a sparse-kernel implementation in which only the unmasked parameters participate in backpropagation. This substantially reduces the per-step computation and memory footprint, allowing GMixout to run on limited hardware. This is why initially we reported 9.0M for GMixout and 85.5M for Mixout
>
> To avoid confusion, we have clarified in the text (L372) accompanying Table 3 that GMixout and Mixout have the same total number of learnable parameters (85.5M). We have also updated the parameter count for Mixout in Table 3 to match GMixout, with an added footnote explaining that the effective per-step parameter count depends on the specific implementation.

---

### Official Review · Reviewer_Tydt · 2025-10-31

**Soundness:** 3
**Presentation:** 3
**Contribution:** 2
**Rating:** 6
**Confidence:** 4

**Summary:**

This paper studies and improves OOD robustness during fine-tuning by 1) revisiting MIXOUT, a stochastic regularization technique that replaces finetuned weights with pretrained references, and 2) proposes GMixout to improve robust finetuning of vision foundation models. The authors first analyze Mixout through bias-variance-covariance-locality (BVCL) decomposition to get an ensemble-based theoretical understanding. Then GMixout is designed with two key method-level modifications: (1) replacing the fixed pretrained anchor with an exponential moving average which can adapt during training and (2) controlling mask resampling frequency with a hyperparameter. Memory and compute efficiency is considered with sparse CUDA kernels. The method is evaluated across diverse distribution shift scenarios and demonstrates consistent improvements in OOD robustness with competitive ID accuracy.

**Strengths:**

- **Insightful initial theoretical analysis and motivation.** The paper studies a timely and important topic - OOD robustness of fine-tuning methods. The ensemble-based perspective on Mixout and the BVCL decomposition provides explanation for why mask sparsity, EMA coefficient, and resampling frequency matter for robustness. Then GMixout focuses on improving robustness with these three points.
- **Comprehensive experimental evaluation and ablation studies.** The paper evaluates on (1) a diverse set of benchmarks including different types of distribution shift (covariate shift, coruptions, long-tail) and (2) many baseline methods (full fine-tuning, PEFTs, and Model Soups), which strengthen the claims’ generalizability. Section 5.3 provides multiple ablation studies of different hyparameters, parameter budgets, and architectural choices (Vision vs. VL). GMixout shows higher OOD robustness while maintaining ID accuracy.
- **Practical efficiency. The authors take memory and computational efficiency into consideration.** The sparse CUDA kernel implementation addresses a limitation of the original Mixout. This helps make the improved method feasible for finetuning large-scale models on consumer GPUs (table 3).

**Weaknesses:**

We thank the authors for submitting the paper to ICLR 2026! There are a few weaknesses listed below which I believe can make the paper better. For some points, please also refer to the questions below.
- **Missing statistical significance testing and inconsistent performance on large-scale data.** Throughout the paper, performance differences are often small (0.3-1 point) without error bars, confidence intervals, or significance tests reported. It makes it hard to tell whether improvements are meaningful or within noise margins. This is especially true for larger-scale data (as acknowledged in Observation 2 and table 4), where GMixout achieves only slightly higher average OOD robustness and lower robustness to some fine-tuning strategies (Model Soups and Random Mask). Also, more analysis and understanding on why GMixout sometimes underperforms would strengthen the contribution.
- **Novel limitation.** GMixout is based on Mixout and the core modification (EMA anchor and resampling frequency) are relatively incremental. EMA-based weight averaging is well-established, and the resampling frequency is hyperparameter to tune. The connection to ensemble methods, which I find interesting to read, tends to be more interpretative than technically novel. But I think the idea of studying and combining all these techniques and the importance of this topic have a strong weight. I think this is not a big concern for me.
- **Incomplete comparison with related work.** The paper does include a relatively thorough evaluation framework, but there are comparisons with a limited number of other recent robust fine-tuning methods beyond Model Soups and basic PEFT.

**Questions:**

- Can you provide error bars across multiple random seeds? Given the margins in many comparisons, this would strengthen the claims considerably.
- Why does GMixout’s advantage narrow on ImageNet-1k compared to medium-scale datasets? Is this a fundamental limitation of the approach or coulter more hyperparameter tuning (episode I) help?
- With long-tail dataset ImageNet-LT (table 2), GMixout achieves the best “few” shot accuracy and is behind Model Soups on “many” shot. Could you provide insight into this trade-off and whether it can be adjusted?

---

> ### Author Response · Authors · 2025-11-21
> **Response to Reviewer Tydt**
>
> We thank the reviewer for their thoughtful feedback and for highlighting our theoretical motivation, comprehensive evaluation, and practical sparse-kernel implementation. We address their questions below:
>
> > ### Missing statistical significance testing and Incomplete comparison with related work.
>
> Please refer to the ***More comprehensive baselines*** section in the **Global response [2/2]** for full details. Here, we briefly highlight the key points.
>
> Per the reviewers’ request, we expand our baselines to six additional methods and evaluate them on two DomainNet settings (training on Real or Sketch and testing on the remaining domains), as well as on iWildCam. We also evaluate GMixout across multiple random seeds under the same setup. As shown in the table in the global response, **none of the new baselines outperform GMixout**, and GMixout’s variance across seeds is negligible, further supporting the robustness of our method.
>
> > ### Inconsistent performance on large-scale data / Why does GMixout’s advantage narrow on ImageNet-1k compared to medium-scale datasets? Is this a fundamental limitation of the approach or coulter more hyperparameter tuning (episode I) help?
>
> GMixout is most effective in over-parameterized, under-data fine-tuning regimes, where ERM easily overfits and implicit ensembling (via remasking) reduces variance and improves OOD robustness. This explains the larger gains on DomainNet, iWildCam, and CIFAR-100, where data scarcity is a core challenge.
>
> On ImageNet-1k, overfitting is less severe due to the dataset’s scale, so the room for additional variance reduction is naturally smaller. As shown in Figure 2, increasing the amount of ImageNet-1k training data further narrows the gap: GMixout still provides consistent improvements, but with reduced margins.
>
> Overall, GMixout sits in a sweet spot; yielding strong benefits on medium-scale datasets while remaining competitive on large-scale ones such as ImageNet-1k. Although additional hyperparameter tuning (particularly for ImageNet-1k) could likely recover part of this gap, we intentionally keep configurations fixed across datasets for fairness.
>
> > ### Novel limitation. GMixout is based on Mixout and the core modification (EMA anchor and resampling frequency) are relatively incremental.
>
> Please refer to the ***Lack of innovation*** section in the **Global response [1/2]**.
>
> > ### With long-tail dataset ImageNet-LT (table 2), GMixout achieves the best “few” shot accuracy and is behind Model Soups on “many” shot. Could you provide insight into this trade-off and whether it can be adjusted?
>
> It is well established in the class-imbalance literature [1,2] that, on imbalanced datasets, maximizing accuracy on many-shot (head) classes is relatively easy. In fact, standard cross-entropy training, without any imbalance-specific techniques, often achieves near–state-of-the-art performance on head classes. Consequently, the primary challenge in imbalance settings is to maximize accuracy on a balanced test distribution, which requires strong and well-balanced performance across both head and tail classes.
>
> Model Soups tend to favor head-class performance because they fully fine-tune on the original imbalanced distribution, improving head accuracy but harming tail accuracy. In contrast, GMixout updates only a small subset of parameters within each iteration, which slightly reduces head-class gains but preserves more balanced representations, ultimately yielding better tail-class generalization.
>
> *[1] Shi, Jiang-Xin, et al. "Long-tail learning with foundation model: Heavy fine-tuning hurts." ICMLR 2024.*
>
> *[2] Cao, Kaidi, et al. "Learning imbalanced datasets with label-distribution-aware margin loss." Neurips 2019.*

---

### Official Review · Reviewer_rUAF · 2025-11-01

**Soundness:** 3
**Presentation:** 4
**Contribution:** 2
**Rating:** 4
**Confidence:** 4

**Summary:**

This paper addresses the trade-off between in-domain performance and robustness when finetuning vision foundation models. The authors revisit Mixout, interpreting it as an implicit ensemble via stochastic weight-sharing, and identify three key robustness factors: anchor choice, resampling frequency, and sparsity. They propose GMixout, which adapts the anchor using an exponential moving average and introduces an explicit resampling-frequency hyperparameter. A sparse-kernel implementation ensures efficiency with no inference overhead. Across benchmarks including ImageNet, DomainNet, iWildCam, and CIFAR100-C, GMixout improves finetuning accuracy while outperforming model soups and parameter-efficient baselines under distribution shift.

**Strengths:**

1. The paper is clearly written, well structured, and visually polished.
2. The experimental evaluation is extensive and convincingly supports the paper’s claims across diverse benchmarks.

**Weaknesses:**

1. The comparison is incomplete — several strong robust finetuning methods that dynamically constrain weight drift from pretrained models, such as TPGM[1], FTP[2], SPD[3], are not included.
2. The evaluation is limited to vision models; no experiments are provided on large language or vision-language models, which would strengthen the generality of the proposed approach.

[1] Tian, Junjiao, et al. "Trainable projected gradient method for robust fine-tuning." Proceedings of the IEEE/CVF Conference on Computer Vision and Pattern Recognition. 2023.

[2] Tian, Junjiao, et al. "Fast trainable projection for robust fine-tuning." Advances in Neural Information Processing Systems 36 (2023): 11374-11393.

[3] Tian, Junjiao, Chengyue Huang, and Zsolt Kira. "Rethinking weight decay for robust fine-tuning of foundation models." Advances in Neural Information Processing Systems 37 (2024): 22418-22440.

**Questions:**

Please refer to the weaknesses.

---

> ### Author Response · Authors · 2025-11-21
> **Response to Reviewer rUAF**
>
> We thank the reviewer for their thoughtful feedback and for highlighting the clarity of our presentation and the strength of our experimental evaluation. We address their questions below:
>
> > ### W1. The comparison is incomplete, several strong robust finetuning methods that dynamically constrain weight drift from pretrained models
>
> Per the reviewers’ request, we incorporated AdamSPD [1], the most recent dynamic weight-decay approach among the suggested methods, into our evaluation on the two DomainNet settings (training on Real or Sketch and testing on the remaining domains) as well as on iWildCam. The results are summarized in the table below.
>
>
> | Method                 | DN Real       | DN Sketch      | iWildCam        |
> |------------------------|---------------|----------------|-----------------|
> | Model Soups            | 64.3          | 65.9           | **37.6**        |
> | LoRA                   | 66.5          | 71.0           | 35.9            |
> | Random Mask            | 66.8          | 72.0           | 35.2            |
> | AdamSPD                | 66.2          | 68.5           | 36.1            |
> | GMixout                | **68.4±0.19** | **75.6±0.31**  | **37.7±0.73**   |
>
> **GMixout outperforms AdamSPD by a clear margin across all benchmarks.** On DomainNet, PEFT methods remain strong baselines because their inherent constraints on weight drift [2] help stabilize adaptation under moderate distribution shift. In contrast, on iWildCam, where the data distribution is much farther from CLIP’s pretraining domain, AdamSPD gains a slight advantage over PEFT due to its ability to dynamically adjust weight penalties. Nevertheless, GMixout consistently achieves the best performance across all evaluated settings.
>
> *[1] Tian, Junjiao, Chengyue Huang, and Zsolt Kira. "Rethinking weight decay for robust fine-tuning of foundation models." Neurips (2024).*
>
> *[2] Biderman, Dan, et al. "Lora learns less and forgets less." TMLR (2024).*
>
> > ### W2. The evaluation is limited to vision models; no experiments are provided on large language or vision-language models
>
> Please refer to the ***Extension to language*** section in the **Global response [1/2]**.

---

### Official Review · Reviewer_WqZb · 2025-11-01

**Soundness:** 3
**Presentation:** 3
**Contribution:** 3
**Rating:** 4
**Confidence:** 3

**Summary:**

This paper revisits Mixout through the lens of implicit ensemble regularization, conceptually similar to stochastic regularization such as Dropout.
By doing this, this paper proposes GMixout, a generalized variant that improves supervised fine-tuning (SFT) OOD robustness under distribution shift.
The authors argue that Mixout’s random parameter replacement implicitly forms an ensemble of subnetworks sharing the same backbone weights, and that controlling this stochastic process can enhance out-of-distribution (OOD) generalization.
A sparse CUDA implementation further enables scaling to large ViT/CLIP backbones.
Reported image classification experiments across various benchmarks demonstrate consistent OOD robustness improvements compared to Mixout and other baseline methods, without sacrificing in-distribution (ID) accuracy.

**Strengths:**

This method empirically improves the OOD robustness.
The authors evaluate across multiple OOD settings and achieves superior OOD robustness in most settings.
he sparse CUDA implementation makes Mixout-style regularization feasible on modern large-scale vision models, demonstrating fair engineering contribution.

**Weaknesses:**

The method is effective but not very well motivated. Where does the improvement come from, why the adaptive anchor and the resampling frequence method is effective.  It will add to great value if more in-depth analysis could be made about the `implicit ensemble` feature of the GMixout process, instead of just describing them intuitively.

GMixout primarily modifies Mixout via some hyperparameters (EMA anchor and resampling frequency). While effective, it may be viewed as an engineering refinement rather than a conceptual breakthrough.

All reported experiments are conducted on vision classification tasks (ViT, CLIP).
However, the SFT-based algorithms, especially those PEFT methods, have also been primarily deployed to those more advanced models and/or tasks. For example, it is expected to report GMixout performance on LLM, VLM, or even those reasoning models, to justify its universal effectiveness and efficiency.

**Questions:**

Why model soup performs better on Cifar100?
The results in Table 4 seems not to be favor to the proposed method.

---

> ### Author Response · Authors · 2025-11-21
> **Response to Reviewer WqZb**
>
> We thank the reviewer for their thoughtful feedback and for recognizing our implicit-ensemble perspective, the GMixout formulation, and the practical sparse-CUDA implementation. We address their questions below:
>
> > ### W1. The method is effective but not very well motivated. Where does the improvement come from, why the adaptive anchor and the resampling frequence method is effective. It will add to great value if more in-depth analysis could be made about the implicit ensemble feature of the GMixout process, instead of just describing them intuitively.
>
> Section 4.1 outlines the intuition behind GMixout’s effectiveness using the BVCL perspective. Figures 1, 3, and 4 further demonstrate how the resampling strategy, mask sparsity, and moving anchor jointly shape the ID–OOD trade-off. To deepen this analysis, we added a new study in Appendix A.7 (highlighted in blue) that quantifies prediction diversity across GMixout subnetworks, providing additional evidence of its implicit ensembling behavior.
>
> Prediction diversity measures subnetwork diversity using the ratio-error metric, computed across all checkpoint pairs from GMixout runs on DomainNet-Sketch with $\lambda \in \{0.1,0.5,0.9\}$ and $k \in \{1,100\}$. Higher values mean subnetworks tend not to make the same mistakes. We find that:
>
> - with $k=1$, prediction diversity is virtually zero for most checkpoint pairs; the subnetworks behave almost identically;
> - increasing $k$ markedly boosts diversity;
> - for fixed $k$, diversity is maximized at $\lambda=0.1$ and decreases as $\lambda$ increases.
>
> Overall, GMixout enhances robustness by producing a more diverse set of hypotheses through frequent resampling.
>
>
> > ### W2. GMixout primarily modifies Mixout via some hyperparameters (EMA anchor and resampling frequency). While effective, it may be viewed as an engineering refinement rather than a conceptual breakthrough.
>
> Please refer to the ***Lack of innovation*** section in the **Global response [1/2]**.
>
> > ### W3. GMixout performance on LLM, VLM, or even those reasoning models, to justify its universal effectiveness and efficiency.
>
> Please refer to the ***Extension to language*** section in the **Global response [1/2]**.
>
> > ### Q1. Why model soup performs better on Cifar100?
>
> Please note that Model Soups are often treated as an impractical upper bound, as they require training and storing multiple full models, making them computationally expensive [1]. In contrast, GMixout surpasses the remaining baselines by large margins (gap of 3.4), and approaches the performance of model soups (gap of 0.9), while updating only a single model and only a fraction of its parameters at each iteration.
>
> *[1] Jang, Dong-Hwan, Sangdoo Yun, and Dongyoon Han. "Model stock: All we need is just a few fine-tuned models." ECCV 2024.*

---

### Official Review · Reviewer_EpdY · 2025-11-01

**Soundness:** 3
**Presentation:** 2
**Contribution:** 2
**Rating:** 4
**Confidence:** 3

**Summary:**

This paper revisits Mixout, a stochastic regularizer for finetuning pretrained models, and reinterprets it as a single run implicit ensemble mechanism. Based on this new perspective, the authors identify three core factors influencing robustness：1、Masking anchor：the reference weights that Mixout reverts to. 2、Resampling frequency：how often random masks are refreshed. 3、Mask sparsity：the proportion of weights replaced or retained. Building on this, the paper proposes GMixout, which (i) replaces the fixed pretrained anchor with an exponential moving average (EMA) of weights during training, and (ii) introduces a resampling frequency hyperparameter that controls how frequently subnetworks are resampled.The authors also present a sparse kernel GPU implementation that enables large scale finetuning on consumer GPUs without inference time cost. Extensive experiments on benchmarks such as ImageNet, DomainNet, iWildCam, CIFAR100 C, and ImageNet LT show that GMixout improves out of distribution (OOD) robustness while maintaining or improving in domain (ID) accuracy compared with LoRA, Random Mask, and Model Soups.

**Strengths:**

The work provides a novel theoretical reinterpretation of Mixout as an implicit ensemble in weight space；The proposed EMA based adaptive anchor and mask resampling frequency control are conceptually simple yet innovative extensions that directly improve robustness；The analysis using bias variance covariance–locality (BVCL) decomposition is insightful, linking ensemble theory to parameter efficient finetuning (PEFT).The experiments are comprehensive and rigorous, covering covariate shift, corruption, and class imbalance.Ablation studies (Figure 3，4) systematically examine how EMA coefficient, resampling frequency, and sparsity affect IDOOD trade offs.Results are consistent across multiple datasets and model sizes, showing strong empirical support.The insights may inspire further exploration of ensemble theoretic views of other finetuning methods (e.g., LoRA or adapters).

**Weaknesses:**

Limited exploration on language or multimodal tasks：Although the authors claim GMixout is general, all experiments are on vision datasets. Demonstrating its applicability on language or vision–language tasks would reinforce generality.

Comparison with newer PEFT baselines：The baselines include LoRA and Random Masking, but recent adapter free PEFT approaches (e.g., DoRA, AdaLoRA, and QLoRA) are not discussed empirically. Including would provide stronger positioning. Although the ensemble based bias，variance，covariance，locality analysis is conceptually appealing, the theoretical derivation is mostly heuristic. A more formal proof or tighter bounds on the expected OOD error under Mixout/GMixout would strengthen the argument.

**Questions:**

1. refer to weakness

2. Can GMixout be applied to text based or multimodal foundation models (e.g., CLIP, LLaVA, or language- only transformers)? If so, are there any expected differences in behavior? and how sensitive is performance to λ and k when scaling to larger models like ViT-L/14 or other architectures (e.g., ResNet backbones)?

---

> ### Author Response · Authors · 2025-11-21
> **Response to Reviewer EpdY [1/2]**
>
> We thank the reviewer for their thoughtful feedback and for highlighting our implicit-ensemble view of Mixout, the GMixout design, and the breadth of our empirical evaluation. We address their questions below:
>
> > ### W1. Limited exploration on language or multimodal tasks
>
> Please refer to the ***Extension to language*** section in the **Global response [1/2]**.
>
> > ### W2.1 Comparison with newer PEFT baselines.
>
> Per the reviewers’ request, we incorporated DoRA [1] ,the most recent alternative among the suggested LoRA variants, into our evaluation on the two DomainNet settings (training on Real or Sketch and testing on the remaining domains) as well as on iWildCam. The results are summarized in the table below.
>
>
> | Method                 | DN Real       | DN Sketch      | iWildCam        |
> |------------------------|---------------|----------------|-----------------|
> | LoRA                   | 66.5          | 71.0           | 35.9            |
> | Random Mask            | 66.8          | 72.0           | 35.2            |
> | DoRA                   | 66.0          | 70.6           | 35.7            |
> | GMixout (Ours)         | **68.4±0.19** | **75.6±0.31**  | **37.7±0.73**   |
>
> **GMixout outperforms DoRA by a clear margin across all benchmarks**. Interestingly, Random Mask, as recently observed in [2], continues to be a strong PEFT baseline, yet GMixout still achieves the best results in every evaluated setting.
>
> *[1] Liu, Shih-Yang, et al. "Dora: Weight-decomposed low-rank adaptation." ICML 2024.*
>
> *[2] Xu, Jing, and Jingzhao Zhang. "Random masking finds winning tickets for parameter efficient fine-tuning." ICML 2024.*
>
> > ### W2.2 Although the ensemble based bias，variance，covariance，locality analysis is conceptually appealing, the theoretical derivation is mostly heuristic. A more formal proof or tighter bounds on the expected OOD error under Mixout/GMixout would strengthen the argument.
>
> We appreciate the request for stronger theoretical guarantees. Our aim, however, is not to provide distribution-agnostic theory, but to offer a practical framework that guides the design of GMixout. The BVCL perspective serves as an interpretive lens that yields testable predictions along three control axes, anchor source, resampling frequency, and mask sparsity. Figures 1, 3, and 4 already analyze how GMixout’s resampling strategy, mask sparsity and moving anchor shape the ID–OOD trade-off. To further examine its implicit ensembling behavior, we added a new analysis in Appendix A.7 (highlighted in blue) that measures prediction diversity across GMixout subnetworks.
>
> Prediction diversity measures subnetwork diversity using the ratio-error metric, computed across all checkpoint pairs from GMixout runs on DomainNet-Sketch with $\lambda \in \{0.1,0.5,0.9\}$ and $k \in \{1,100\}$. Higher values mean subnetworks tend not to make the same mistakes. We find that:
>
> - with $k=1$, prediction diversity is virtually zero for most checkpoint pairs; the subnetworks behave almost identically;
> - increasing $k$ markedly boosts diversity;
> - for fixed $k$, diversity is maximized at $\lambda=0.1$ and decreases as $\lambda$ increases.
>
> Overall, GMixout enhances robustness by producing a more diverse set of hypotheses through frequent resampling.

---

> ### Author Response · Authors · 2025-11-21
> **Response to Reviewer EpdY [2/2]**
>
> > ### Q1. Can GMixout be applied to text based or multimodal foundation models?
>
> GMixout makes no assumptions that restrict it to the vision domain. We have also begun running experiments in the language domain and will make every effort to include preliminary results before the end of the discussion period. Please refer to the ***Extension to language*** section in the global response for full details.
>
> > ### Q2.1 How sensitive is performance to $\lambda$ and $k$ when scaling to larger models like ViT-L/14 or other architectures (e.g., ResNet backbones)?
>
> Due to computational constraints, our ablation studies were conducted on ViT-B/16 variants. We then transferred the best hyperparameters to ViT-L/14 for the results shown in Table 5. That said, based on both our observations and the underlying mechanics of GMixout, we expect the following trends to hold as model size increases:
>
> 1. We expect $k$ to follow the same trend observed in ViT-B/16: increasing $k$ promotes lower correlation among subnetworks, thereby improving the implicit ensemble effect. Larger models generally have more representational capacity, so this effect may become even more pronounced.
> 2. The optimal value of $\lambda$ is more sensitive to the baseline performance of the pretrained backbone. Since larger models (e.g., ViT-L/14) typically start from stronger pretrained performance, we expect the optimal $\lambda$ to shift upward. Intuitively, better pretrained features allow more aggressive mixing before degrading OOD performance.
>
> We will make every effort to include additional results along these lines by the camera-ready deadline.
>
> > ### Q2.2 GMiout for other architectures (e.g., ResNet backbones)?
>
> We believe there is no fundamental limitation preventing GMixout from being applied to convolutional architectures. The main consideration is handling the spatial correlations in convolutional feature maps, which may require a masking procedure tailored to their structure [1].
>
> *[1] Ghiasi, Golnaz, Tsung-Yi Lin, and Quoc V. Le. "Dropblock: A regularization method for convolutional networks." Neurips 2018.*

---

### Author Response · Authors · 2025-11-21
**Global response [1/2]**

We thank all reviewers for their constructive feedback. In this global response, we address three concerns that were shared by the majority of reviewers. We then respond to each reviewer’s specific comments in their corresponding sections.

## Lack of innovation

We emphasize that our contribution goes well beyond simply adding EMA anchoring and resampling frequency to Mixout. Specifically, our work provides:

**1. A new analytical lens for understanding Mixout:** To the best of our knowledge, we are the first to analyze Mixout through the lens of the BVCL decomposition [3]. Since Mixout implicitly defines a weight-space ensemble, this decomposition offers a principled way to understand why Mixout underperforms in many OOD settings. Our analysis (Fig 1,3,4) show that, unlike effective ensembles, Mixout fails to generate sufficiently decorrelated subnetworks. During the rebuttal period, we extended our analysis with a prediction-diversity study (Appendix A.7 highlighted in blue), which confirms that Mixout subnetworks exhibit almost no diversity, providing a clear explanation for its consistent failure modes.

**2. GMixout: A principled generalization of Mixout:** Guided by these insights, we propose GMixout, a structured and theoretically motivated adaptation of Mixout. GMixout preserves the core idea of weight mixing while introducing three key mechanisms that directly target the limitations identified in our BVCL and diversity analyses:

- Resampling frequency: a new hyperparameter controlling how often subnetworks are refreshed. Together with mask sparsity, this increases subnetwork decorrelation and enables Mixout to recover true ensemble benefits.
- EMA anchor: replacing the fixed pre-trained weights used in Mixout with a moving-average anchor. This provides smoother optimization trajectories and adapts the model more effectively to OOD data.
- Sparse CUDA kernels: an efficient implementation that updates only the active parameters, making GMixout genuinely memory- and compute-efficient on consumer GPUs.

**3. Comprehensive empirical validation:** We empirically validate the above insights with extensive ablations and cross-domain evaluations. GMixout consistently outperforms strong baselines across several challenging OOD scenarios, including:

- Covariate shift: ImageNet (Tab 4), DomainNet (Tab 1), iWildCam (Tab 1)
- Common corruptions: CIFAR100-C (Tab 1)
- Class imbalance: ImageNet-LT (Tab 2), CIFAR100-LT (Tab 2)

## Extension to language

Although GMixout makes no assumptions that restrict it to the vision domain, our experiments focused on vision tasks because this area offers widely studied distribution-shift benchmarks [1,3,7] that fit our computational constraints. We have already begun running language-domain experiments and will make every effort to include preliminary results before the end of the discussion period. However, comprehensive evaluations on language and vision–language tasks (e.g., VQA, image captioning) were beyond our computational budget and are deferred to future work.

Because distribution shifts can arise for many reasons, our evaluation accounts for a broad range of scenarios. Following this principle, and given our computational budget, we prioritized the vision domain, as it offers a rich and diverse testbed for studying robustness. In particular, we evaluate GMixout against strong baselines under **three major shift types** (corruptions, covariate shift, and long-tailed imbalance), across **both medium-scale datasets** (DomainNet, iWildCam, CIFAR-100) and a **large-scale dataset** (ImageNet-1k). These benchmarks also allow us to test GMixout on datasets **close to the CLIP pre-training distribution** (ImageNet, DomainNet) as well as datasets that are **distributionally far from CLIP** (iWildCam, CIFAR100).

---

*[1] Wortsman, et al. 2022, "Model soups: averaging weights of multiple fine-tuned models improves accuracy without increasing inference time."*

*[3] Rame et al. 2022, "Diverse weight averaging for out-of-distribution generalization."*

*[7] Gulrajani, et al. 2020, "In search of lost domain generalization."*

---

> ### Author Response · Authors · 2025-11-21
> **Global response [2/2]**
>
> ## More comprehensive baselines
>
> We selected baselines that are widely recognized for delivering state-of-the-art performance under specific types of distribution shift. In particular, for corruption and covariate shift benchmarks, especially when the target distribution deviates substantially from the pretraining data or when the dataset becomes large, Model Soups [1] is a strong baseline [2,3]. Many prior works [4,5,6] explicitly acknowledge this and aim to match Model Soups’ robustness while reducing computational cost.
>
> However, following the reviewers’ request for broader comparisons, we additionally evaluated:
>
> 1. PEFT method (**DoRA**)[8],
> 2. General robust fine-tuning methods (**LP-FT**[10] and **AdamSPD**[9]), and
> 3. Recent robust fine-tuning methods specifically designed for vision-language models (**CLIPOOD**[12], **CaRot**[13], **FLYP**[12]).
> 4. We also include **WiseFT**[14], a general model-averaging baseline previously detailed in Appendix A.2 (under WISE-FT extension).
>
> We evaluate all methods on two DomainNet settings (trained on Real or Sketch and evaluated on the remaining domains) as well as on iWildCam. Furthermore, we run GMixout with three different random seeds to demonstrate its robustness to stochastic variation. The table below summarizes these extended results alongside the performance of our main baselines.
>
>
> | Method                 | DN Real       | DN Sketch      | iWildCam        |
> |------------------------|---------------|----------------|-----------------|
> | Zero-shot              | 65.8          | 72.8           | 12.7            |
> | Model Soups            | 64.3          | 65.9           | **37.6**        |
> | LoRA                   | 66.5          | 71.0           | 35.9            |
> | Random Mask            | 66.8          | 72.0           | 35.2            |
> | WiseFT                 | **68.9**      | 75.2           | 31.1            |
> | **New Baselines**      |               |                |                 |
> | LP-FT                  | 59.3          | 61.2           | 31.8            |
> | DoRA                   | 66.0          | 70.6           | 35.7            |
> | AdamSPD                | 66.2          | 68.5           | 36.1            |
> | CLIPOOD                | 60.2          | 62.6           | 24.7            |
> | CaRot                  | 67.8          | 73.7           | -               |
> | FLYP                   | 57.9          | 63.9           | -               |
> | **Ours**               |               |                |                 |
> | GMixout                | **68.4±0.19** | **75.6±0.31**  | **37.7±0.73**   |
>
>
> As shown in the table, **none of the newly added baselines surpass GMixout**. The only two methods that come close are:
>
> 1. CaRot, which performs competitively but requires full-parameter training of both the vision and language towers, leading to substantially higher computational and memory cost. In contrast, GMixout delivers strong robustness while activating only 10% of the parameters, offering a far more efficient trade-off.
> 2. WiseFT is a general weight-averaging procedure that, as shown in Table 7 (Appendix A.2), can be applied to all methods, including GMixout. However, its effectiveness is highly dataset-dependent. On DomainNet, where the zero-shot model is already strong, WiseFT improves all baselines, particularly on OOD performance. In contrast, on iWildCam, where CLIP’s zero-shot accuracy is weak, WiseFT offers no benefit; GMixout in fact performs better without it. Overall, GMixout delivers consistent OOD gains across both datasets, regardless of the quality of the underlying zero-shot CLIP initialization.
>
> ---
>
> *[1] Wortsman, et al. 2022, "Model soups: averaging weights of multiple fine-tuned models improves accuracy without increasing inference time."*
>
> *[2] Fort, et al. 2019, "Deep ensembles: A loss landscape perspective."*
>
> *[3] Rame et al. 2022, "Diverse weight averaging for out-of-distribution generalization."*
>
> *[4] Jang, et al. 2024, "Model stock: All we need is just a few fine-tuned models."*
>
> *[5] Garipov, et al. 2023, "Loss surfaces, mode connectivity, and fast ensembling of dnns."*
>
> *[6] Rame, et al. 2023, "Model ratatouille: Recycling diverse models for out-of-distribution generalization."*
>
> *[7] Gulrajani, et al. 2020, "In search of lost domain generalization."*
>
> *[8] Liu, et al. 2024, "Dora: Weight-decomposed low-rank adaptation."*
>
> *[9] Tian, et al. 2024, "Rethinking weight decay for robust fine-tuning of foundation models."*
>
> *[10] Kumar, et al. 2022, "Fine-tuning can distort pretrained features and underperform out-of-distribution."*
>
> *[11] Shu, et al. 2023, "Clipood: Generalizing clip to out-of-distributions." ICML 2023.*
>
> *[12] Goyal, et al. 2023, "Finetune like you pretrain: Improved finetuning of zero-shot vision models."*
>
> *[13] Zhu, et al. 2023, "Robust fine-tuning of zero-shot models via variance reduction."*
>
> *[14] Wortsman, et al. 2022, "Robust fine-tuning of zero-shot models."*

---

### Author Response · Authors · 2025-11-27
**Language understanding experiment**

For language understanding, we evaluate on CivilComments [1], a standard benchmark for studying subpopulation shift [2,3]. Each comment is annotated with a binary toxicity label and an 8-dimensional binary vector indicating whether it references any of eight demographic identities: male, female, LGBTQ, Christian, Muslim, other religions, Black, and White. This setting reflects an imbalance-recognition problem in which the main labels follow a skewed distribution, while the relevant attribute driving distribution shift (the demographic identity) is unknown during training. As a result, models tend to overfit to majority groups and exhibit poor performance on minority subpopulations. We follow the standard WILDS benchmark splits [2], and demographic labels remain unavailable throughout training.

Following the methodology in [3], we fine-tune BERT$_{\text{BASE}}$ and report worst-group accuracy, defined as the minimum accuracy across the demographic subpopulations. This metric directly captures robustness to distributional imbalance, a core challenge in CivilComments given its pronounced subpopulation shift. We apply PEFT layers to all of the Linear layers within transformers.

The table below summarizes the results for six fine-tuning strategies, including GMixout.


| **Method**         | **Worst acc.** |
|--------------------|------------|
| Full-FT            | 63.6       |
| Model Soups        | 62.5       |
| LoRA               | 58.5       |
| Random Mask        | 56.8       |
| Mixout             | 67.6       |
| DoRA               | 64.5       |
| **GMixout (Ours)** | **69.0**   |

GMixout achieves the highest worst-group accuracy (69.0%), outperforming all baselines.

---

*[1] Borkan, Daniel, et al. "Nuanced metrics for measuring unintended bias with real data for text classification." WWWC 2019.*

*[2] Marklund, et al. "Wilds: A benchmark of in-the-wild distribution shifts." ICML 2021.*

*[3] Yang, Yuzhe, et al. "Change is hard: A closer look at subpopulation shift." ICML 2023.*

---

### Author Response · Authors · 2025-11-30
**Authors Final Remarks**

We thank all reviewers for their constructive feedback. We have updated the manuscript accordingly. In particular:

1. We incorporated the new prediction-diversity analysis (Figure 3), providing additional evidence of GMixout’s implicit ensembling behavior.
2. We extended our evaluation to a language-understanding task (Table 6), further demonstrating GMixout’s effectiveness under distribution shift in the language modality.
3. We evaluated and compared GMixout against six additional baseline methods on two DomainNet settings (trained on Real or Sketch and evaluated on the remaining domains) as well as on iWildCam. Due to space constraints, these new baseline results are reported in the Appendix (Table 8). For the camera-ready version, we plan to extend these evaluations to the full DomainNet and CIFAR100-C datasets and update Table 1 accordingly.
4. We evaluated GMixout under three seeds on both DomainNet settings and on iWildCam to demonstrate the statistical significance of the results (Table 8).
5. We further clarify the number of learnable parameters in Mixout and GMixout, as well as the impact of sparse-kernel implementations on their efficiency (Table 3).
6. We updated the related-work section to reflect the new benchmarks and analyses.

We have highlighted all new changes in blue in the revised manuscript.

During the rebuttal, Reviewer 26yR (original rating: 2) noted that, after reviewing the newly added results and comparisons with SOTA robust fine-tuning methods, their impression of the paper [“shifted significantly toward the positive side.”](https://openreview.net/forum?id=UiATTlpNLz&noteId=mYtDOE0SS9)

We thank the AC for their time and consideration.

---

### Meta-Review · Area_Chair_wV5H · 2026-01-07

**Summary:**

This AC carefully independently reviewed the paper  and summarized the main concerns along with questions:

* Reviewer EpdY - limited exploration on language or multimodal tasks beyond vision datasets; missing comparisons with recent PEFT baselines.

* Reviewer WqZb - While the method is effective, it lacks clear motivations and intuitions for the proposed improvements and designs; GMixout appears to modify Mixout through hyperparameter choices, and all reported experiments are limited to vision classification tasks.

* Reviewer rUAF - incomplete comparisons and evaluation restricted to vision models; no experiments on language or vision language models.

* Reviewer Tydt - incremental novelty over Mixout; incomplete comparisons with related work; missing statistical significance testing; inconsistent performance on large-scale data.

* Reviewer 26yR - lack of technical or theoretical innovation; limited baselines for comparison; and missing many references.

**Reviewer Concerns:**

Some concerns about limited evaluation were partially addressed by the authors' additional experiments provided during the rebuttal period, which include the language understanding experiments reported in Table 6 and the expanded comparisons and citations of prior work in Table 8. The AC appreciates the authors' efforts for additional experiments. However, extending the evaluation beyond the vision modality would require more comprehensive studies, which should be particularly beyond a single language modeling dataset. It is also understood that conducting such comprehensive evaluations under the authors' time constraints is often infeasible; however, performing them on a single dataset may not constitute a sufficient contribution, as the original experiments were perceived as insufficient. As a result, the reviewers' concerns for limited evaluation have been only partially addressed.

Furthermore, most reviewers share the concern that the technical novelty of the proposed method over the baseline Mixout is limited, and the AC concurs with this assessment. While the authors' claim that this work is the first to analyze Mixout through the lens of the BVCL decomposition is reasonable, the method itself appears to constitute an incremental improvement over Mixout rather than a substantial methodological advance. The AC speculates that the novelty concern, along with the limited evaluation concerns, may stem from the fact that Mixout originally focused on the language modality rather than vision.

Specifically, as noted by Reviewer WqZb, the approach primarily modifies Mixout through additional hyperparameters, such as the EMA anchor and resampling frequency. Reviewer Tydt similarly observes that GMixout is fundamentally based on Mixout, with its core modifications being relatively incremental. Furthermore, as pointed out by Reviewer 26yR, the idea of moving or updating the anchor has been explored in prior work on robust fine-tuning (e.g., Jang et al., 2024), and the benefits of exponential moving average (EMA) during fine-tuning have also been well studied in existing methods (e.g., Shu et al., 2023; Oh et al., 2024).

Finally, multiple reviewers raise overlapping concerns on the limited novelty of the proposed method, which the AC considers critical. Therefore, the AC believes the paper should make stronger contributions beyond the baseline and encourages the authors to further enhance the method's intrinsic novelty in future revisions.

**Reviewer Scores:**

Reviewer 26yR indicated that their rating could be increased following the rebuttal and actively engaged in the discussion. Unfortunately, the other reviewers did not participate in the discussion, and based on the remaining concerns outlined above, the AC believes that their scores are unlikely to improve.

---

### Decision · Program_Chairs · 2026-01-26

Reject